# Apical PAR complex proteins protect against programmed epithelial assaults to create a continuous and functional intestinal lumen

Maria Danielle Sallee, Melissa A Pickett, Jessica L Feldman*

Department of Biology, Stanford University, Stanford, United States

**Abstract** Sustained polarity and adhesion of epithelial cells is essential for the protection of our organs and bodies, and this epithelial integrity emerges during organ development amidst numerous programmed morphogenetic assaults. Using the developing *Caenorhabditis elegans* intestine as an in vivo model, we investigated how epithelia maintain their integrity through cell division and elongation to build a functional tube. Live imaging revealed that apical PAR complex proteins PAR-6/Par6 and PKC-3/aPkc remained apical during mitosis while apical microtubules and microtubule-organizing center (MTOC) proteins were transiently removed. Intestine-specific depletion of PAR-6, PKC-3, and the aPkc regulator CDC-42/Cdc42 caused persistent gaps in the apical MTOC as well as in other apical and junctional proteins after cell division and in non-dividing cells that elongated. Upon hatching, gaps coincided with luminal constrictions that blocked food, and larvae arrested and died. Thus, the apical PAR complex maintains apical and junctional continuity to construct a functional intestinal tube.

## Introduction

Epithelia are dynamic tissues composed of polarized and adherent epithelial cells that line organs, provide mechanical resiliency, act as selective barriers, and separate animal bodies from the outside world. The integrity of a functional epithelium requires cell polarity and adhesion to be consistent and continuous throughout the tissue (*Pickett et al., 2019*). Epithelial cells polarize along an apico-basal axis, collectively positioning their apical surfaces facing outward with important specializations such as microvilli and cilia, while their basolateral domains contact neighboring cells and basement membranes. Junctional complexes ('junctions') form belt-like bands at the cell periphery that adhere neighboring cells together and separate the apical and basolateral domains. During development and homeostasis, epithelia must maintain their integrity despite being challenged by various programmed cellular 'assaults' such as mitosis, cell shape change, intercalation, extrusion, and cell death, any of which could potentially compromise tissue architecture and function. How epithelial primordia maintain their integrity in vivo in the face of these challenges to build a functional organ is not well understood.

Different assaults on epithelial integrity require individual epithelial cells to remodel existing polarized structures or build new ones. In most epithelial cells, cell division physically cleaves their polarized domains and daughters must build junctions at the new cell-cell contact while maintaining correct apicobasal polarity (*Le Bras and Le Borgne, 2014*). Division also requires rearrangement of the microtubule cytoskeleton. In many interphase epithelial cells, such as *Caenorhabditis elegans* and mammalian intestinal cells, *Drosophila* tracheal and follicle cells, and many mammalian epithelial cell lines, microtubules are arranged in parallel arrays emanating from an apically localized microtubule-organizing center (MTOC), allowing for intracellular vesicle transport and apicobasal polarity

*For correspondence:
feldmanj@stanford.edu

Competing interests: The authors declare that no competing interests exist.

maintenance (*Sanchez and Feldman, 2017*). During mitosis, microtubules are temporarily rearranged into radial arrays around the centrosomal MTOC and then returned to the apical surface (*Pease and Tirnauer, 2011*; *Yang and Feldman, 2015*), yet how microtubules are released from and returned to the apical surface is unknown. Epithelial cells frequently must also change their shape and size. For example, the epithelial folds that drive *Drosophila* gastrulation and vertebrate neurulation require cells to undergo apical construction, shrinking their apical surfaces in concert with their junctions (*Martin and Goldstein, 2014*); conversely, new cells are added to the *Xenopus* epidermis via apical emergence, expanding their apical surfaces and junctions as they extend to join the apical side of the epithelium (*Sedzinski et al., 2016*).

The highly conserved apical PAR polarity complex can play important roles in epithelial cell remodeling by, for example, orienting the mitotic spindle during mitosis (*Kuchinke et al., 1998*; *Mack and Georgiou, 2014*; *Hao et al., 2010*; *Siegrist and Doe, 2005*) and regulating actin dynamics during apical constriction (*David et al., 2010*; *Ishiuchi and Takeichi, 2011*; *Walck-Shannon et al., 2016*; *Zilberman et al., 2017*). This 'PAR complex,' consisting of PAR-3/Par3, PAR-6/Par6, PKC-3/aPkc, and CDC-42/Cdc42, is best characterized for its role in the formation and maintenance of apicobasal domains and junctions in many tissues and organisms (*Pickett et al., 2019*). The scaffold Par6 binds the kinase aPkc; another scaffold Par3 recruits them to the apical surface, and the Rho-family GTPase Cdc42 replaces Par3 and activates aPkc kinase activity (*Achilleos et al., 2010*; *Harris and Peifer, 2005*; *Hutterer et al., 2004*; *Joberty et al., 2000*; *Kim et al., 2009*; *Lin et al., 2000*; *Rodriguez et al., 2017*; *Totong et al., 2007*). aPkc phosphorylates and removes basolateral proteins from the apical domain, and mutual inhibition between apical and basolateral proteins maintains and refines the apical and basolateral domains (*Pickett et al., 2019*). The PAR complex also promotes adherens junction plasticity, tight junction formation, and junction maintenance (*Georgiou et al., 2008*; *Montoyo-Rosario et al., 2020*; *Rusu and Georgiou, 2020*; *Totong et al., 2007*; *Wallace et al., 2010*; *Zilberman et al., 2017*). These roles are well known, though specific contributions can vary by tissue (*Pickett et al., 2019*). One example is in the early steps of *C. elegans* embryonic intestinal development. PAR-3 is required to establish the apical domain of intestinal cells along a shared central midline, which later forms the lumen, and to recruit other PAR complex proteins and microtubules to the apical surface (*Achilleos et al., 2010*; *Feldman and Priess, 2012*); PAR-6 and PKC-3 localize interdependently to the apical surface, and PAR-6 is required for junction formation but not polarity (*Montoyo-Rosario et al., 2020*; *Totong et al., 2007*); and CDC-42 is not required for polarity or junction formation (*Zilberman et al., 2017*). However, epithelial remodeling occurs after polarity establishment as intestinal cells divide and elongate, and the role of PAR complex proteins in these later stages is less well understood due to their earlier requirements for embryo viability (*Achilleos et al., 2010*; *Gotta et al., 2001*; *Kay and Hunter, 2001*; *Montoyo-Rosario et al., 2020*; *Plusa et al., 2005*; *Tabuse et al., 1998*; *Totong et al., 2007*; *Watts et al., 1996*; *Zilberman et al., 2017*). Recent advances in tissue-specific protein depletion allow robust spatial and temporal control of protein removal, enabling the study of the later roles of polarity proteins in specific tissues (*Armenti et al., 2014*; *Sallee et al., 2018*; *Wang et al., 2017*; *Zhang et al., 2015*).

We used the developing *C. elegans* intestine as a genetically accessible in vivo context to study how a polarized epithelial primordium maintains tissue integrity through the developmental assaults of cell division and elongation to build a functional tube. We found that PAR complex proteins are required during embryonic intestinal development to build a functional larval intestine with a continuous, open lumen. Intestine-specific depletion of PAR-6, PKC-3, and CDC-42 in embryos resulted in gaps along the intestinal midline in apical MTOC proteins, both where cells had divided and where cells were elongating. Preventing polarized cells from dividing in PAR-6-depleted intestines also reduced their tendency to form gaps, suggesting a causal link between division and gap formation. In PAR-6-depleted intestines, gaps in MTOC proteins correlated with gaps in apical and junctional proteins, and basolateral proteins failed to be excluded from the midline. The discontinuities in apical and junctional proteins became more numerous as elongation progressed, with PAR-6 continuing to be required in later stages of embryonic intestinal elongation, and larvae arrested upon hatching with functionally obstructed intestines showing gaps in the lumen, apical surfaces, and junctions. Together, these results demonstrate the importance of these highly conserved PAR complex proteins during development in remodeling the apical domain and junctions to maintain continuity as intestinal cells divide and expand their apical surfaces.

## Results

### Cells within the polarized *C. elegans* intestine elongate and divide

Following polarization, cells in the developing *C. elegans* intestine divide, intercalate, and elongate during morphogenesis, and thus serve as a good in vivo model to understand the mechanisms epithelial cells use to buffer against these assaults. The 16-cell primordial intestine (E16) is arranged into two tiers of 10 dorsal and 6 ventral cells. These cells begin to polarize soon after they are born, building a continuous apical domain along a central midline, the future lumen (*Leung et al., 1999*). Adherens and septate-like junctions form a single 'apical junction' that localizes around the apical domain between left/right, dorsal/ventral, and anterior/posterior pairs of cells in a ladder-like arrangement along the midline (*Costa et al., 1998*; *Labouesse, 2006*; *Leung et al., 1999*; *Figure 1—figure supplement 1A, B*). Two pairs of anterior and posterior cells, referred to here as 'star cells,' undergo an additional round of mitosis after polarity is established while the other 12 'non-star cells' remain in interphase (*Figure 1—figure supplement 1C*). The embryonic intestine more than doubles its midline length from a polarized primordium at bean stage to the 1.5- to 1.8-fold stage (*Figure 1A–C*). Elongation continues and the lumen is formed at the midline. Embryos hatch as L1 (first larval stage) larvae with a functional intestine (*Figure 1C*), built from a series of nine stacked 'int' rings with a continuous central hollow lumen through which food passes (*Figure 1—figure supplement 1A*). The apical surfaces of all the cells face the lumen where they secrete digestive enzymes and absorb nutrients through microvilli, and junctional complexes maintain cell adhesion and barrier function while allowing the open lumen to run unobstructed between the intestinal rings (*Dimov and Maduro, 2019*).

### PAR complex proteins remain apical during mitosis when the apical MTOC is transiently removed

We first examined the localization of different polarized proteins during mitosis to determine the impact of cell division on their localization. Apically organized microtubules are one hallmark of epithelial cells that is necessarily affected by mitosis. Apical microtubules and the microtubule nucleator γ-Tubulin Ring Complex (γ-TuRC) transiently leave the apical surface when star cells divide and centrosomes are reactivated as MTOCs to build the mitotic spindle (*Pease and Tirnauer, 2011*; *Yang and Feldman, 2015*). The loss of apical microtubules is easiest to see when star cells divide at the same time. Both apical surfaces lose apical microtubules during mitosis, creating a visible 'midline gap' in MTOC proteins as both the left and right cells reactivate centrosomal MTOCs; the midline gap is then refilled upon mitotic exit (*Figure 1D, E*, *Figure 1—figure supplement 2A–C, F*; *Yang and Feldman, 2015*). In addition to microtubules and the γ-TuRC components TBG-1/γ-tubulin and GIP-1/GCP3, all other MTOC and microtubule-associated proteins we examined also left the apical surface during mitosis (*Figure 1E*). Some MTOC proteins localized to the active centrosomes (γ-TuRC, ZYG-9/chTOG, AIR-1/Aurora A), while other proteins did not and likely became cytoplasmic (NOCA-1/Ninein, PTRN-1/CAMSAP, VAB-10B/Spectraplakin). The observed removal of apical MTOC proteins during mitosis, including the MTOC anchoring proteins VAB-10B and WDR-62 (*Sanchez et al., 2020*), suggests the complete inactivation of the apical surface as an MTOC. Junctional proteins retained some localization during mitosis, albeit reduced as compared with interphase neighboring cells (*Figure 1F*). Localization of the septate-like junction-associated protein DLG-1/Discs large was most strongly reduced during mitosis (*Figure 1—figure supplement 2B, D*), and the adherens junction protein HMR-1/E-cadherin was also reduced compared to the non-dividing 'non-star' intestinal cells (*Figure 1—figure supplement 2B, E*), appearing as thin fluorescent threads as new cell boundaries formed. The other apical proteins examined, ACT-5/actin and three members of the PAR complex PAR-3/Par3, PAR-6/Par6, and PKC-3/aPkc, all remained apically localized during mitosis (*Figure 1G, H*, *Figure 1—figure supplement 2C, G*). Therefore, we hypothesized that PAR complex proteins actively mark the apical surface during mitotic remodeling to return apical MTOC function after division, thereby restoring a continuous apical MTOC.

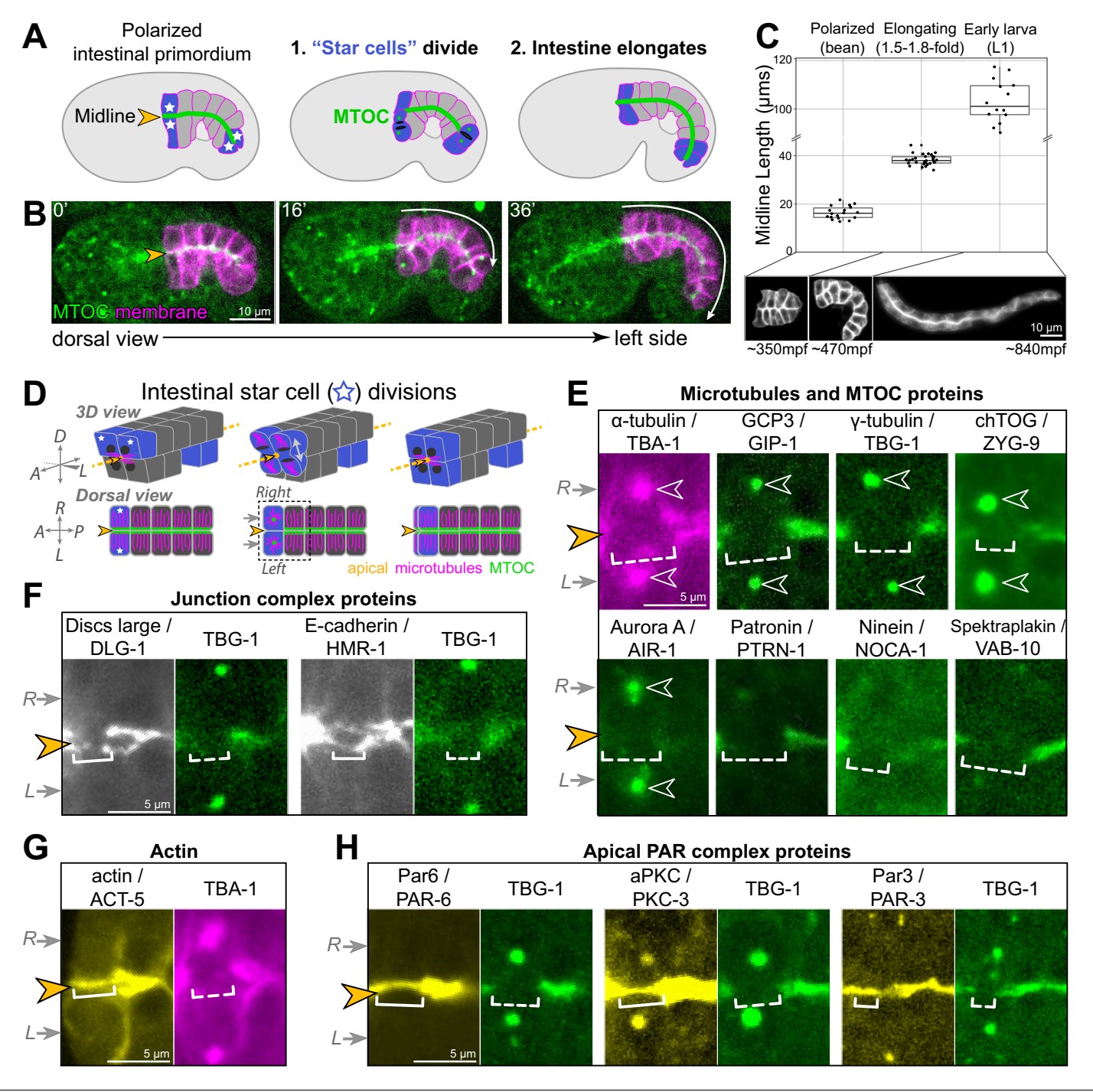

**Figure 1.** Cell division and elongation challenge the epithelial integrity of the developing *C. elegans* intestine. (**A, B**) A cartoon schematic (**A**) and corresponding live time course (**B**) of an embryo expressing an apical microtubule-organizing center (MTOC) marker TBG-1::mCherry (green, orange arrowhead) and a membrane marker intestinal GFP::CAAX (magenta); the anterior and posterior 'star cells' divide (blue) and the apical surface elongates (white arrow) as the polarized intestinal primordium develops into an intestine. (**C**) Top: graph showing intestinal apical length in newly polarized primordia (bean stage, average length = 16.5 ± 2.6 µm, n = 18), ~1.5–2 hr after the start of intestinal elongation (1.5-fold to 1.8-fold, average length = 38.2 ± 2.2 µm, n = 30), and upon hatching in the L1 larval stage (average length = 105.0 ± 11.1 µm, n = 15). Below: corresponding intestinal GFP::CAAX images and approximate age, in minutes post-fertilization (mpf). (**D**) Cartoon schematic of the anterior star cell divisions illustrating their dorsoventral division in 3D (top) and from a dorsal view (bottom). Dotted line indicates viewing angle of images in (**E–H**). (**E–H**) Live imaging of indicated proteins relative to TBG-1/γ-tubulin::mCherry (green) or mCherry::TBA-1/α-tubulin (magenta) and the midline (orange arrowhead) when the right and left cells divided synchronously (gray arrows). Note that in some cases a midline gap (dashed bracket) formed, and in other cases no gap

*Figure 1 continued on next page*

*Figure 1 continued*

formed (solid bracket). Numbers of embryos with midline gaps in mitotic star cells, assessed by eye: mCherry::TBA-1 (5/5), ZF::GFP::GIP-1 (5/5), TBG-1:: mCherry (10/10), mCherry::AIR-1 (10/10), ZYG-9::ZF::GFP (3/3), PTRN-1::GFP (5/5), NOCA-1::ZF::GFP (7/7), VAB-10B::GFP (3/3), HMR-1::GFP (0/8), DLG-1::mNG (1/8), YFP::ACT-5 (0/6), PAR-6::GFP (0/10), GFP::PKC-3 (0/5), and PAR-3::GFP (0/5). All images are maximum intensity Z-projections (0.5–1.5 µm) that capture centrosomes (open white arrowheads) and/or the intestinal midline. Scale bar = 10 µm in (B, C). Scale bar = 5 µm in (E–H).

The online version of this article includes the following figure supplement(s) for figure 1:

**Figure supplement 1.** *C.elegans* intestinal development.

**Figure supplement 2.** Protein localization before and during star cell divisions.

## PAR-6 and PKC-3 are required for the return of apical MTOC function after mitosis

To test our hypothesis that the PAR complex acts as an apical memory mark, we asked if removing PAR-6 or its well-characterized binding partner PKC-3 during the star cell divisions would disrupt the return of the MTOC to the apical surface. To avoid the embryonic lethality caused by loss of PAR-6 or PKC-3 in the zygote or in the skin (*Montoyo-Rosario et al., 2020*; *Tabuse et al., 1998*; *Totong et al., 2007*; *Watts et al., 1996*), we used the ZIF-1/ZF degradation system to deplete each protein only in the intestine ('gut(-),' *Armenti et al., 2014*; *Sallee et al., 2018*). Briefly, we used CRISPR/Cas9 to add a ZF::GFP tag to endogenous PKC-3 and PAR-6; the ZF1 degron ('ZF') facilitated efficient degradation by the E3 ligase component ZIF-1, and GFP allowed visualization of protein localization and depletion. We used intestine-specific promoters to express ZIF-1 prior to polarity establishment (*elt-2*p in the four-intestinal cell 'E4' stage and *ifb-2*p in the eight-intestinal cell 'E8' stage), ensuring robust depletion only in the gut well before star cell divisions and intestinal elongation (*Figure 2—figure supplement 1*). Unlike PAR-3, PAR-6 and PKC-3 are not required to establish apicobasal polarity in the intestine (*Achilleos et al., 2010*; *Totong et al., 2007*; *Figure 2—figure supplement 1*), so we could degrade PAR-6 in early stages of intestinal development without disrupting polarity establishment itself and ensure robust depletion prior to the star cell divisions.

In control embryos that had completed their star cell divisions (late comma- to 1.8-fold stage), we observed continuous midline localization of the apical MTOC markers TBG-1::mCherry and PTRN-1:: GFP (*Figure 2A*). In contrast, we observed gaps in the apical MTOC in PAR-6$^{gut(-)}$ and PKC-3$^{gut(-)}$ embryos (*Figure 2B, C*). Gaps were often observed toward the anterior and posterior ends of intestines, the approximate position of star cell daughters (*Figure 2D*). To further test if gaps formed where star cells had divided, we used live time-lapse imaging of the MTOC in dividing star cells. No gaps were observed during asynchronous star cell divisions in control and PAR-6$^{gut(-)}$ embryos (*Figure 2—figure supplement 2*). In control embryos in which anterior star cells divided synchronously, a midline gap formed in the apical MTOC as centrosomes recruited additional TBG-1; MTOC proteins returned and filled the gap within 12 min as the centrosomes inactivated and shed TBG-1 (*Figure 2E, F*). In PAR-6$^{gut(-)}$ embryos, however, midline MTOC gaps that formed during synchronous divisions persisted long after mitotic exit through the end of the time course, for a minimum of 24 min (*Figure 2E, G*). The persistence of the midline gap in MTOC proteins in PAR-6$^{gut(-)}$ and PKC-3$^{gut(-)}$ embryos is consistent with our hypothesis that PAR complex proteins promote the return of apical MTOC function after division.

## PAR-6 and PKC-3 are required during elongation to maintain apical MTOC continuity

In addition to the challenge that mitosis poses to the apical continuity of a polarized epithelium, the cell growth and shape changes underlying intestinal morphogenesis could similarly interrupt epithelial integrity. Intestinal length increases approximately 70% in the hour between the star cell divisions and 1.5-fold stage (from 22.4 ± 3.1 µm to 38.2 ± 2.2 µm), so we asked if PAR-6 and PKC-3 are required for the cell shape changes that both the star and non-star cells undergo during intestinal elongation. We first assessed the continuity of the apical MTOC during elongation using TBG-1:: mCherry and PTRN-1::GFP as markers. In elongating control intestines, the apical MTOC remained continuous along the midline in both star cell daughters and in non-star cells (*Figure 3A*, inset). A line scan tracing the midline of control intestines showed some variation in apical MTOC protein intensity but no apparent gaps in signal intensity (*Figure 3A′*). By contrast, in elongating PAR-6$^{gut(-)}$

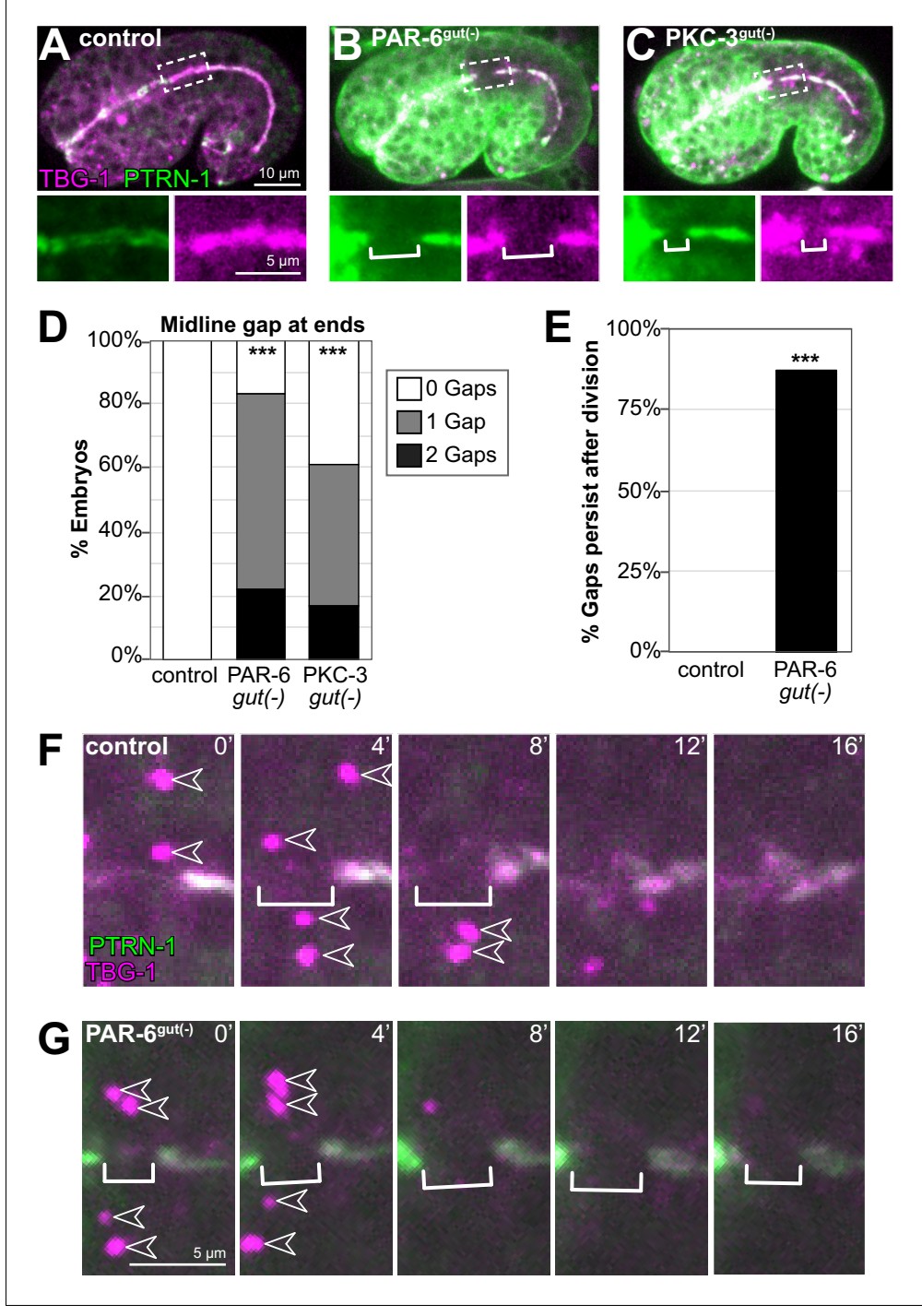

**Figure 2.** PAR-6 and PKC-3 are required for the return of microtubule-organizing center (MTOC) function to the apical surface following mitosis. (A–C) Dorsolateral images of live comma- to 1.5-fold-stage embryos of indicated genotypes expressing transgenic MTOC markers TBG-1::mCherry (magenta) and endogenous PTRN-1::GFP (green). Maximum intensity Z-projections (1–3 μm) capture the intestinal midline. Enlarged images of boxed regions shown below. Star cell divisions had completed, but gaps (white bracket) were observed in PAR-6gut(-) and PKC-3gut(-) embryos. (B, C) Strong GFP fluorescence outside the intestine is undegraded PAR-6::ZF::GFP and ZF:: GFP::PKC-3 in non-intestinal tissues. (D) Graph showing the percent of embryos with midline gaps in TBG-1 and PTRN-1 at the anterior and/or posterior regions of the intestine, approximately where star cell divisions occurred. Control: n = 22, PAR-6gut(-): n = 18, PKC-3gut(-): n = 18. Statistical analysis: ANOVA with Tukey's post hoc tests. Differences from control indicated with asterisks. (E) Graph showing the percent of control (n = 0/5) and PAR-6gut(-)

*Figure 2 continued on next page*

*Figure 2 continued*

(n = 7/8) embryos in which a midline gap in TBG-1::mCherry and PTRN-1::GFP formed in synchronously dividing cells and persisted through the end of the 32 min time-lapse movie, with the gap itself persisting for at least 24 min. Statistical analysis: Fisher's exact test. In control embryos, the gap lasted 8 min or less, except for one which lasted 12 min. (F, G) Representative examples from (E) of live time-lapse imaging of TBG-1 and PTRN-1 during star cell divisions in indicated genotypes. Brackets mark the apical MTOC gap; arrowheads indicate active centrosomal MTOCs marked with TBG-1 localization. All experiments used *ifb-2*p::*zif-1* to drive E8 onset of degradation. Scale bars = 10 μm in (A), 5 μm in (A) inset and (F). ***p<0.001.

The online version of this article includes the following figure supplement(s) for figure 2:

**Figure supplement 1.** ZIF-1/ZF-mediated protein degradation.
**Figure supplement 2.** Asynchronous star cell divisions in control and PAR-6 embryos do not cause gaps in the apical microtubule-organizing center (MTOC).

and PKC-3$^{gut(-)}$ intestines, midline gaps in MTOC proteins formed both in star cell daughters and in non-star cells that had not recently divided (*Figure 3B, C*, insets); the midline gaps were clearly visualized by line scans (*Figure 3B', C'*). Consistent with elongation causing gaps to form, we observed only few PAR-6$^{gut(-)}$ embryos with small midline gaps in TBG-1 and PTRN-1 prior to the star cell divisions and elongation (control: n = 1/17, PAR-6$^{gut(-)}$: n = 5/29). Using time-lapse imaging, we observed gaps in the apical MTOC emerge over time in individual embryos during elongation. In control intestines, the apical MTOC stayed continuous as the average midline length increased approximately 80% in 50 min (*Figure 3D*). In PAR-6$^{gut(-)}$ intestines, however, the apical MTOC formed gaps as the average apical surface elongated 70% in 50 min (*Figure 3E*).

To quantify the apical MTOC defects in PAR-6$^{gut(-)}$ and PKC-3$^{gut(-)}$ embryos, we defined a 'background-level' measurement as a midline intensity measurement for which both TBG-1::mCherry and PTRN-1::GFP signal intensities were in the same range as intestinal cytoplasmic background signal (see Materials and methods), and a 'gap' as three consecutive midline intensity measurements that were background level. For this analysis, we included both star and non-star cell gaps. By these stringent criteria, 55.5% of PAR-6$^{gut(-)}$ and 44.4% of PKC-3$^{gut(-)}$ embryos had one or more gaps compared with 0% of control embryos (*Figure 3F*). As a total proportion of the midline MTOC line scan signal, we found that 4.1% of PAR-6$^{gut(-)}$ and 3.7% of PKC-3$^{gut(-)}$ midline measurements were background level compared to 0% in control intestines (*Figure 3G*). The above criteria likely underestimate gap frequency as more gaps were observed by eye (compare *Figure 3F* to *Figure 4E*). Finally, to determine if more gaps arose during elongation, we measured and compared MTOC protein signal intensity at the midline just after the anterior divisions complete (t = 10') and after 50 min of elongation (t = 60'). The number of background-level measurements increased over time in PAR-6$^{gut(-)}$ embryos but not in control embryos (*Figure 3H*). The increased number of gaps in PAR-6$^{gut(-)}$ embryos was likely not due to additional cell divisions as we did not observe active centrosomes with TBG-1::mCherry, and intestines generally had 20 cells (n = 22/23); one embryo had 21 intestinal cells, as is occasionally observed in wild-type embryos (*Sulston and Horvitz, 1977*), but it had no midline gaps. These results suggest that in addition to their roles in promoting continuity in dividing cells, PAR-6 and PKC-3 are also required for maintaining apical MTOC continuity between neighboring non-dividing cells during intestinal elongation.

## Midline gaps form across cell-cell interfaces often due to cell division

Previous studies demonstrated that PAR-6 is essential for correct junction formation in the intestine (*Totong et al., 2007*). To determine where gaps form and whether they result from cell separation due to failed adhesion, we depleted PAR complex proteins PAR-6, PKC-3, and the Rho GTPase CDC-42 using TBG-1::mCherry to mark midline gaps and an intestine-specific GFP::CAAX to label cell membranes. We did not detect physical separation between intestinal cells in control, PAR-6$^{gut(-)}$, PKC-3$^{gut(-)}$, or CDC-42$^{gut(-)}$ embryos (*Figure 4A–F*); the membrane marker clearly spanned the gaps in TBG-1 (*Figure 4B–D* insets), and neighboring intestinal cells remained adjacent to each other.

With intestinal cell membranes marked, we could also map which cells formed gaps within the intestine, and we observed a higher frequency of gap formation in the 8 star cells than in the 12 non-star cells in all three PAR complex depletion backgrounds (*Figure 4F*). To understand if star cell

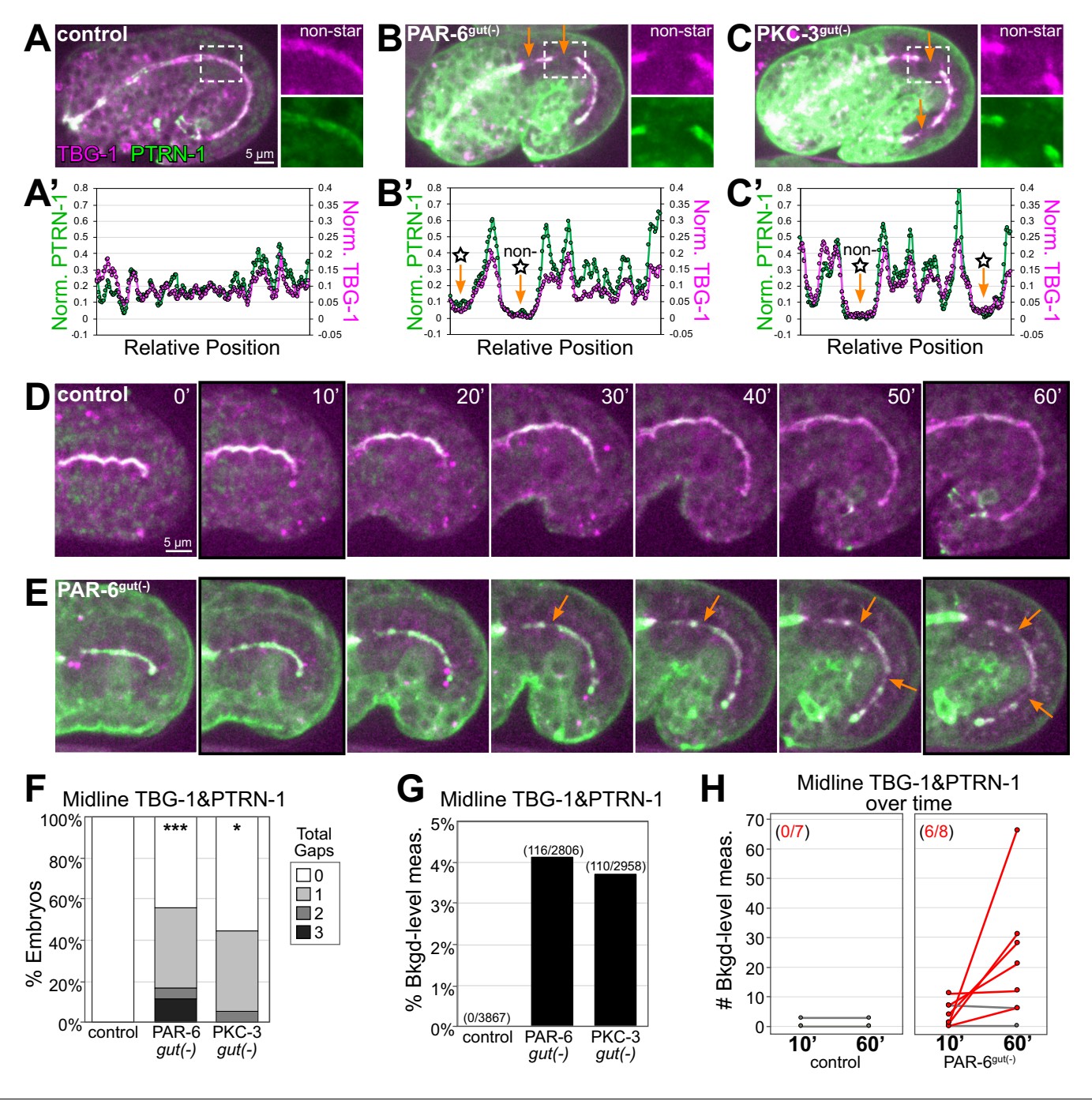

**Figure 3.** PAR-6 and PKC-3 are required to maintain a continuous apical microtubule-organizing center (MTOC) during intestinal elongation. (A–C) Lateral live images of 1.5-fold to 1.8-fold-stage embryos of indicated genotypes expressing MTOC markers TBG-1::mCherry and PTRN-1::GFP. Maximum intensity Z-projections (2.5–3.5 μm) capture the intestinal midline. 2× magnified images of boxed region highlighting the apical MTOC in non-star cells are shown at right. Orange arrows indicate midline gaps. (A'–C') Line scan along the apical midline of corresponding embryos above, plotting midline length-normalized PTRN-1 or TBG-1 signal intensity from anterior to posterior. (D, E) Dorsal-dorsolateral view of posterior half of live embryos developing over 1 hr. Apical MTOC gaps are indicated (orange arrow). t = 0', last frame of anterior star cell mitosis. (F) Graph showing percent of embryos with midline gaps in MTOC proteins as defined by the normalized signal intensity of both TBG-1 and PTRN-1 (see Materials and methods). Control: n = 22, PAR-6$^{gut(-)}$: n = 18, PKC-3$^{gut(-)}$: n = 18. Statistical analysis: ANOVA with Tukey's post hoc tests. Differences from control indicated with asterisks. No significant difference between PAR-6$^{gut(-)}$ and PKC-3$^{gut(-)}$. (G) Graph showing percent of midline intensity measurements in (F) that are 'background level.' Control: 0.00%, PAR-6$^{gut(-)}$: 4.13%, PKC-3$^{gut(-)}$: 3.72%. (H) Graph showing paired comparisons per embryo in the number of background-level measurements at 10 min and 60 min after anterior star cell mitosis. Control: 0/7 embryos increased their number of background-level

*Figure 3 continued on next page*

*Figure 3 continued*

reads; PAR-6$^{gut(-)}$: 6/8 embryos increased their number of background-level reads (red lines). All experiments used *ifb-2*p::*zif-1* to drive E8 onset of degradation. Scale bars = 5 μm. *p<0.05, ***p<0.001.

gaps formed because of mitosis or because of another position-related factor, we blocked the star cell divisions with the cell cycle mutant *cdc-25.2(0)* (*Lee et al., 2016*, *Figure 4G–I*) and asked if fewer star cell gaps formed (see Materials and methods). *cdc-25.2(0)* embryos had 14–16 intestinal cells with no midline gaps (n = 12), and most PAR-6$^{gut(-)}$; *cdc-25.2(0)* embryos had one or more gaps (*Figure 4E*). While midline gaps were more frequent in PAR-6$^{gut(-)}$ star cells than non-star cells (star: 31% vs. non-star: 15%, *Figure 4I*), this difference in gap frequency was abrogated in PAR-6$^{gut(-)}$; *cdc-25.2(0)* double mutant embryos (star: 8% vs. non-star:19%, *Figure 4I*). This result suggests that PAR-6 is specifically required to protect MTOC continuity from disruption by cell division.

## Midline gaps in the apical MTOC also lack apical and junctional proteins and fail to exclude basolateral proteins

To determine if the midline gaps in the MTOC were specific to MTOC proteins or if they reflected more general discontinuities in the apical domain, we examined the localization of other apical markers. In PAR-6$^{gut(-)}$ embryos, apical YFP::ACT-5/actin formed gaps along the midline in both star cell daughters and in non-star cells (10/19 and 9/19 gaps, respectively; *Figure 5A, B*), similar to MTOC protein gaps (*Figure 4F*). In addition, we found midline gaps in PAR-3::tagRFP, which normally localizes to the apical surface and junctions in the intestine (*Achilleos et al., 2010*; *Figure 1— figure supplement 1B*), that overlapped with gaps in PTRN-1::GFP (*Figure 5C, F*). These results suggest that PAR-6 is essential not only for a continuous apical MTOC but also for the general continuity of the apical domain during elongation.

Our analysis of GFP::CAAX in PAR-6$^{gut(-)}$ embryos revealed that 92% of TBG-1 gaps spanned the interface between adjacent cells (*Figure 4F*). We hypothesized that junctional or basolateral proteins might be inappropriately invading and spreading into the apical surface, creating the observed midline gaps in the apical domain. To test this hypothesis, we examined the localization of two apical junction complex proteins, HMR-1/E-cadherin and DLG-1/Dlg (*Bossinger et al., 2001*; *Costa et al., 1998*), relative to gaps in TBG-1::mCherry in PAR-6$^{gut(-)}$ embryos. Normally, HMR-1 and DLG-1 localize as a continuous band at the cell periphery between the apical and basolateral domains and form ladder-like belt junctions at the interfaces between intestinal cells (*Leung et al., 1999*; *McMahon et al., 2001*, *Figure 1—figure supplement 1A, B*). In PAR-6$^{gut(-)}$ embryos, HMR-1::GFP localized to the outer edge of the TBG-1 signal as expected for a junctional protein, but was generally absent from the TBG-1 midline gaps (*Figure 5D, F*). For small TBG-1 gaps, a gap in HMR-1 was sometimes not observed. We also examined DLG-1::mNG localization, which was severely perturbed along the entire midline in PAR-6$^{gut(-)}$ embryos as has previously been reported following PAR-6 depletion (*Totong et al., 2007*). We occasionally found large patches of DLG-1 that colocalized with TBG-1; however, DLG-1 was clearly absent from most TBG-1 gaps (*Figure 5E, F*). Together, these results indicate that junctional complexes do not spread into and fill midline gaps in apical proteins following PAR-6 depletion.

Lastly, we asked whether the basolateral domain inappropriately localized at midline gaps in PAR-6$^{gut(-)}$ embryos. We examined two basolateral markers: LGL-1, which localizes specifically to basolateral domains (*Beatty et al., 2010*), and LET-413/Scribble, which localizes to basolateral and junctional domains (*Legouis et al., 2000*; *McMahon et al., 2001*). Because of a natural twist in the intestine (*Hermann et al., 2000*; *Leung et al., 1999*), the apical domain in anterior cells presented itself as a side view under normal imaging conditions, making it easier to detect the localization of membrane-enriched proteins at the midline of these cells (compare midline GFP::CAAX in anterior and posterior cells of middle image in *Figure 1C*). Therefore, we restricted our analysis of LGL-1 and LET-413 localization to the anterior half of the intestine (int1-4). In control intestines, LGL-1::GFP and LET-413::GFP were observed at lateral surfaces and strongly reduced at the midline where apical markers TBG-1::mCherry and PAR-3::tagRFP localized (*Figure 5G–J*). In PAR-6$^{gut(-)}$ embryos, LGL-1 and LET-413 signal intensities were significantly elevated in most gaps, and similar to intensities at lateral surfaces (*Figure 5G–J*). LGL-1 and LET-413 were also observed at midline 'non-gap' regions,

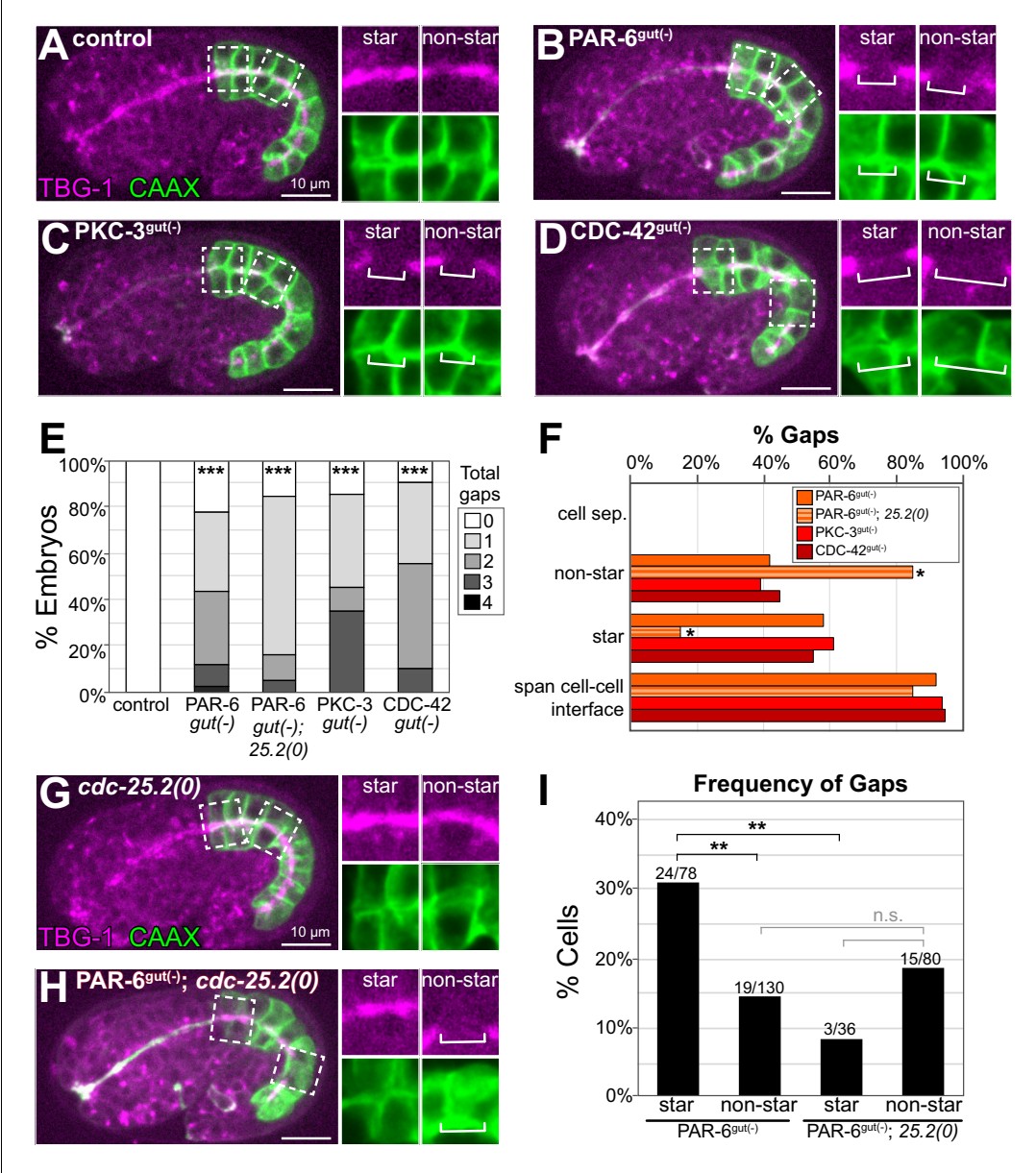

**Figure 4.** PAR-6, PKC-3, and CDC-42 are required for apical microtubule-organizing center (MTOC) continuity but not for cell adhesion. (A–D) Lateral live images of 1.5- to 1.8-fold-stage embryos of indicated genotypes expressing intestine-specific GFP::CAAX and TBG-1::mCherry. Maximum intensity Z-projections (1–2 μm) capture the intestinal midline. 2× magnified images of boxed region highlighting the apical MTOC in star cells (left inset) and non-star cells (right inset). White brackets indicate midline gap. (E) Graph showing percent of embryos with indicated number of TBG-1 gaps assessed by eye. Number of embryos: Control: n = 41, PAR-6$^{gut(-)}$: n = 44, PAR-6$^{gut(-)}$; *cdc-25.2(0)* ('*25.2(0)*'): n = 19, PKC-3$^{gut(-)}$: n = 20, and CDC-42$^{gut(-)}$: n = 20. Statistical analysis: ANOVA with Tukey's post hoc tests. Differences from control indicated with asterisks. (F) Graph showing the percent of gaps of indicated genotypes that corresponded with physical separation of cells ('cell sep.'), that occurred in non-star cells ('non-star') or star cell daughters ('star'), and that spanned the cell-cell interface between anterior/posterior neighbors. Number of gaps: Control: n = 0, PAR-6$^{gut(-)}$: n = 59, PAR-6$^{gut(-)}$; *cdc-25.2(0)*: n = 20, PKC-3$^{gut(-)}$: n = 33, and CDC-42$^{gut(-)}$: n = 31. Statistical analysis: Fisher's exact test comparing the relative proportion of star and non-star cell gaps in PAR-6$^{gut(-)}$ versus PAR-6$^{gut(-)}$; *cdc-25.2(0)* genotypes (see Materials and methods). (G, H) Live lateral images of 1.5-fold-stage embryos of indicated genotypes, markers and insets as in (A–D). The *cdc-25.2(0)* mutation blocks star cell divisions. *cdc-25.2(0)*: n = 12 embryos with no apical TBG-1 gaps. PAR-6$^{gut(-)}$; *cdc-25.2(0)*: n = 19 embryos. (I) Graph showing percent of star or non-star cell interfaces that have a TBG-1::mCherry gap for each genotype (see Materials and methods). PAR-6$^{gut(-)}$: n = 26 embryos, PAR-6$^{gut(-)}$; *cdc-25.2(0)*: n = 19 embryos. Statistical analysis: Fisher's exact test comparing the frequency of gaps in PAR-6$^{gut(-)}$ and PAR-6$^{gut(-)}$; *cdc-25.2(0)* genotypes. See Materials and methods for details of analysis. All experiments used *ifb-2*p::*zif-1* to drive E8 onset of degradation. Scale bars = 10 μm. *p<0.05, **p<0.01, ***p<0.001, n.s. = not significant.

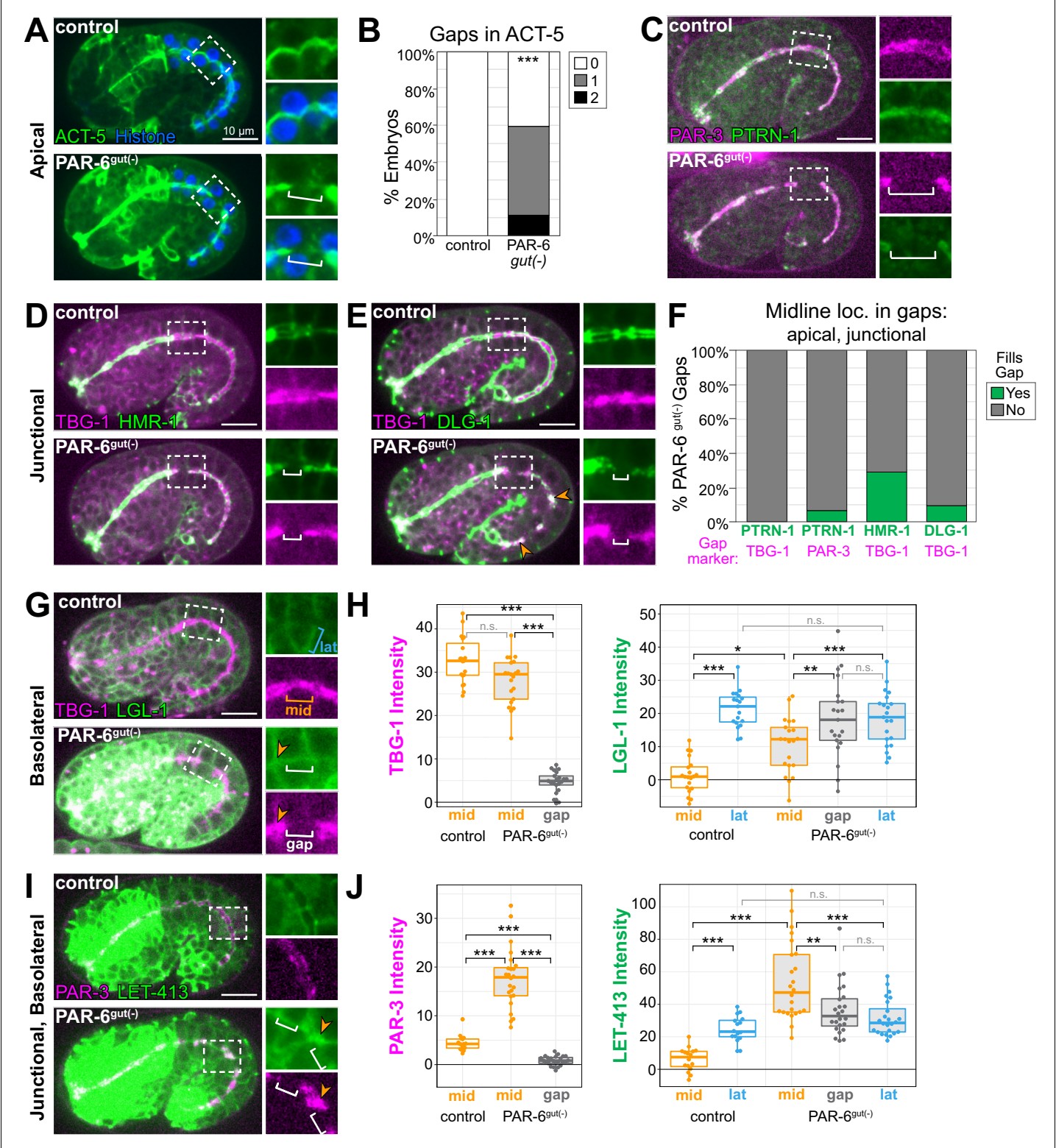

**Figure 5.** Midline gaps in apical and junction proteins overlap and fail to exclude basolateral proteins in PAR-6^gut(-) embryos. (A, C–E, G, I) Lateral live images of 1.5- to 1.8-fold-stage embryos of indicated genotypes expressing indicated markers. Maximum intensity Z-projections (0.5–3.5 μm) capture the intestinal midline, except LGL-1 which is a single Z-slice, and LET-413, which is a minimum intensity Z-projection (1.5–2 μm). 2× magnified images of boxed region highlighting midline gaps are shown at right. White brackets indicate midline gap. Orange arrowheads indicate colocalization of indicated markers. Note that DLG-1::GFP contrast and brightness in (E) was increased in inset PAR-6^gut(-) image to visualize gap. (B) Graph showing

*Figure 5 continued on next page*

**Figure 5 continued**

percent of embryos with indicated number of YFP::ACT-5 gaps. Control: n = 19 embryos, PAR-6$^{gut(-)}$: n = 27. Statistical analysis: Student's t-test. Difference from control indicated with asterisks. (F) Graph showing the percent of midline gaps (magenta) in PAR-6$^{gut(-)}$ embryos to which the indicated protein (green) localized. PTRN-1::GFP/TBG-1::mCherry: n = 0/23 gaps, 18 embryos; PTRN-1::GFP/PAR-3::tagRFP: n = 5/86 gaps, 35 embryos; HMR-1::GFP/TBG-1::mCherry: n = 9/31 gaps, 37 embryos; DLG-1::mNG/TBG-1::mCherry: n = 3/31 gaps, 23 embryos. Microtubule-organizing center (MTOC) gaps were not observed in control par-6(+) embryos (n ≥ 15 embryos per genotype). (H, J) Graphs showing signal intensity at lateral surfaces (blue), the apical midline (orange), and at midline gaps (white) in control and PAR-6$^{gut(-)}$ intestines for indicated markers. LGL-1::GFP/TBG-1::mCherry in PAR-6$^{gut(-)}$: n = 16 anterior gaps, 14 embryos; LGL-1/TBG-1 in controls: n = 0 anterior gaps, 19 embryos; LET-413::GFP/PAR-3::tagRFP in PAR-6$^{gut(-)}$: n = 21 anterior gaps, 15 embryos; LET-413/PAR-3 in controls: n = 0 anterior gaps, 19 embryos. Midline gap localization of LET-413 and LGL-1 was scored only for anterior cells (int1-4). Statistical analysis: Student's t-test with Bonferroni correction. All experiments used ifb-2p::zif-1 to drive E8 onset of degradation except (A) and (B), which used elt-2p::zif-1 to drive degradation at E4. Scale bars = 10 μm. *p<0.05, **p<0.01, ***p<0.001.

where apical markers localized (*Figure 5G, I* insets, orange arrowheads), suggesting a more general failure of LGL-1 and LET-413 exclusion from the apical surface rather than specific invasion of the basolateral domain into the midline gaps. Intriguingly, both PAR-3 and LET-413 signal intensities were highest at non-gap midline regions in PAR-6$^{gut(-)}$ embryos, which could reflect a failure to remove PAR-3 and both junctional and basolateral LET-413 from the midline when PAR-6 is depleted.

## PAR complex-depleted intestinal lumens are functionally obstructed and cannot pass food

The importance of PAR-6 for building a functional intestine is unknown because global loss of par-6 causes lethality in embryos, well before intestinal function can be assessed (*Totong et al., 2007*; *Watts et al., 1996*). Our tissue-specific depletion strategy thus enabled us to bypass par-6 embryonic lethality and examine intestinal morphology and function in PAR-6$^{gut(-)}$ larvae. We hypothesized that defects in apical and junctional continuity in embryonic intestines would lead to morphological and functional defects in the larval intestine. Indeed, all PAR-6$^{gut(-)}$ and PKC-3$^{gut(-)}$ larvae arrested as young L1 larvae (*Figure 6A*, *Figure 6—source data 1*), suggesting that intestinal function was severely compromised. Surprisingly, only 39% of CDC-42$^{gut(-)}$ larvae arrested as L1 larvae.

To better understand the larval arrest caused by intestine-specific loss of these polarity regulators, we asked if intestinal barrier function was disrupted or if the lumen was discontinuous and thus obstructed. We used a 'Smurf' feeding assay to distinguish between these two possibilities (*Gelino et al., 2016*; *Rera et al., 2011*, *Figure 6B, C*, *Figure 6—source data 2*), feeding worms blue-dyed food to monitor whether it leaked into the body (barrier dysfunction) or was trapped in the intestinal lumen (obstructed/discontinuous lumen). In 98% of control larvae, blue food traveled through and filled the entire lumen without leaking into the body (*Figure 6B, D*). However, in 83% of PAR-6$^{gut(-)}$ larvae and 93% of PKC-3$^{gut(-)}$ larvae, blue food passed through the pharynx and pharyngeal valve but was trapped at the anterior end of the intestine, suggesting a luminal obstruction in the first intestinal ring (int1), which comprises the four anterior star cell daughters (*Figure 6B, E, F*). Blue food leaked into the body of a small percentage of larvae (10% of PAR-6$^{gut(-)}$, 2% of PKC-3$^{gut(-)}$), suggesting that most of the PAR-6$^{gut(-)}$ and PKC-3$^{gut(-)}$ larvae that ingested food retained their intestinal barrier function, at least between the pharyngeal valve and int1. Consistent with their milder larval arrest phenotype, CDC-42$^{gut(-)}$ larvae also showed a lower penetrance of obstructed intestines; 42% had a normal intestine (*Figure 6G*), 23% had an early int1 obstruction (*Figure 6G'*), and 20% had a more posterior obstruction.

Using this assay, we also tested whether cell division and elongation contributed to obstructing PAR-6$^{gut(-)}$ intestinal lumens. When the star cell divisions in PAR-6$^{gut(-)}$ embryos were blocked with the cdc-25.2(0) mutation, a larger proportion of resulting larval intestines had a more posterior obstruction than PAR-6$^{gut(-)}$ larvae (25% vs. 5%, *Figure 6B, H*). These results indicate that the defects imposed by the loss of PAR-6 in the dividing star cells could be rescued by blocking the anterior star cell divisions, allowing a continuous int1/int2 connection to form more frequently. To determine if PAR-6 is required during elongation itself and not just during earlier intestinal development, we used a later promoter asp-1 to express intestinal ZIF-1 and degrade PAR-6 in 2-fold-stage embryos ('PAR-6$^{late\ gut(-)}$'), after the star cells divided and before intestinal elongation completed (*Figure 6—figure supplement 1A*). We observed partially penetrant larval arrest (16%, *Figure 6A*) and

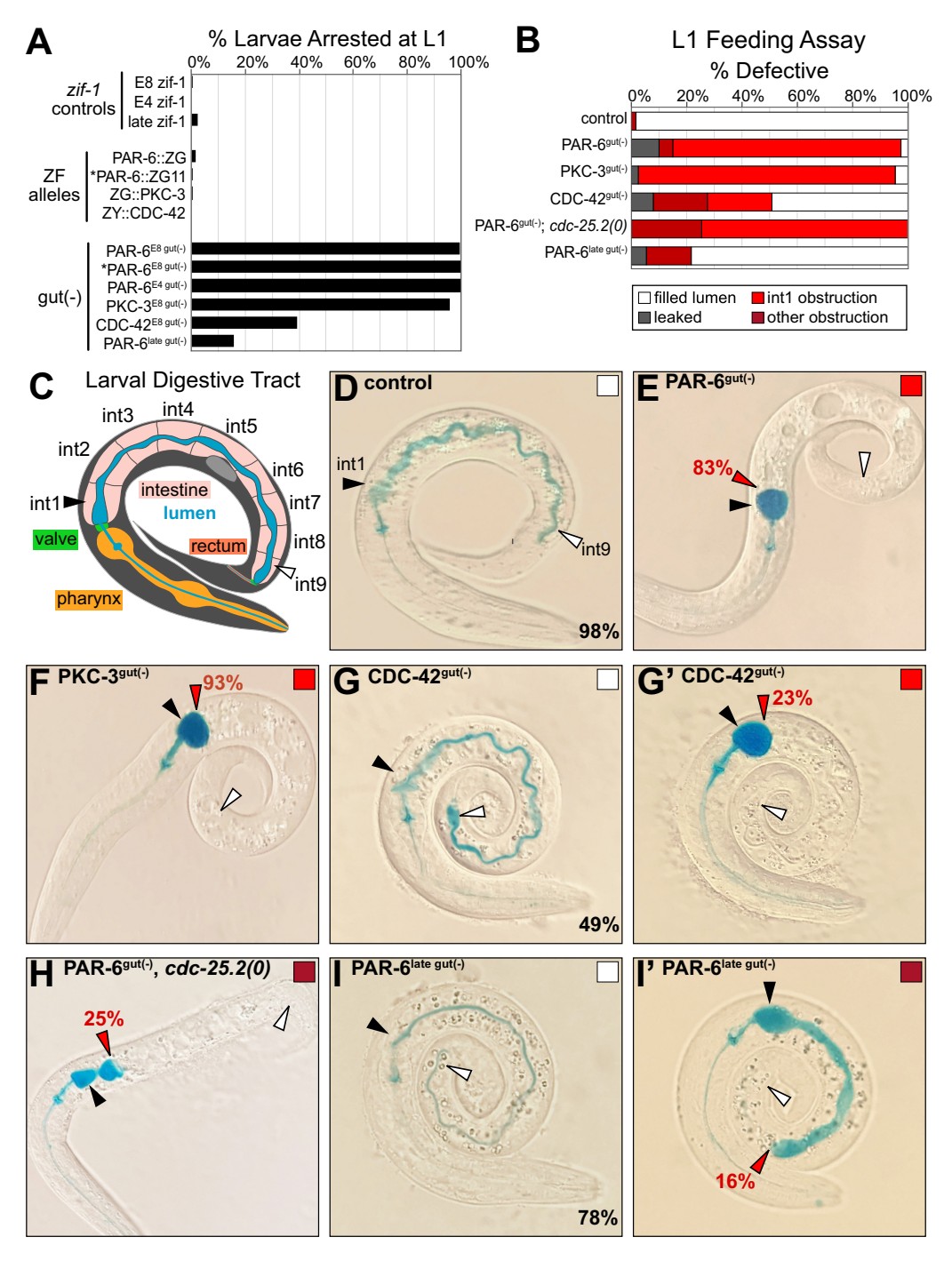

**Figure 6.** PAR-6, PKC-3, and CDC-42 are essential in the embryonic intestine for larval viability and intestinal function. (A) Graph showing the average percent of larvae from three trials that arrested in the L1 stage 72 hr after embryos were laid (see *Figure 6—source data 1*). Genotypes as indicated. Asterisks mark the PAR-6::ZF::GFP$_{11}$ allele (see Materials and methods). (B) Graph showing the average percent of L1 larvae from three trials that had the indicated lumen defects in the feeding assay (see *Figure 6—source data 2*). (C) Cartoon schematic of the larval digestive tract and the path of blue-dyed food in the feeding assay in (D-I'). (D–I') Color images of L1 larvae of indicated genotypes after 3 hr incubation in blue-dyed food. Arrowheads indicate the positions of first intestinal ring int1 (black), the last ring int9 (white), and the anterior-most luminal obstruction (red). Percentages indicate frequency of continuous lumen (D, G, I), int1 obstruction (E, H, I'), and later more posterior obstruction (F, G'). Color-filled square corresponds with feeding assay defect shown. All experiments used *ifb-2*p::*zif-1* to drive E8 onset of degradation except for PAR-6$^{late\ gut(-)}$, which used *asp-1*p::*zif-1* to drive 2-fold-stage onset of degradation, and PAR-6$^{E4\ gut(-)}$, which used *elt-2*p::*zif-1* to drive E4 onset of degradation.

The online version of this article includes the following source data and figure supplement(s) for figure 6:

*Figure 6 continued*

**Source data 1.** Source data for L1 arrest assay in *Figures 6* and *7*.
**Source data 2.** Source data for Smurf feeding assay in *Figures 6* and *7*.
**Figure supplement 1.** Late degradation of PAR-6 in the embryonic intestine.

obstructed lumen phenotypes (16%, *Figure 6B, I, I'*), consistent with a continued requirement for PAR-6 during embryonic intestinal elongation.

## PKC-3 kinase activity and PAR-6-binding are required to build a functional intestine

The interdependent localization of PAR-6 and PKC-3 obscures which of their respective scaffolding and kinase functions is important for maintaining apical continuity and building a functional intestine. To distinguish roles for apical PAR-6 localization and/or PKC-3 kinase activity, we performed a rescue assay with four forms of PKC-3: wild-type PKC-3(+); PKC-3(ΔPB), which removes the PAR-6-binding domain (*Kim et al., 2009*); the canonical K293A kinase-dead mutant PKC-3(K282A); and PKC-3 (G336N), for which the analogous mutation in *Drosophila* aPkc abolishes in vitro kinase activity to the same degree as K293A without losing Par6 binding activity (*Kim et al., 2009*). We found that wild-type PKC-3 was able to rescue the larval arrest and obstructed lumen defects and restore apical PAR-6 localization in embryos (*Figure 7A, E, F*, *Figure 7—figure supplement 1C, G*). As expected, the PAR-6-binding mutant showed almost no larval rescuing activity (*Figure 7B, E, F*), but surprisingly, embryos showed apical PAR-6 enrichment (*Figure 7—figure supplement 1D, G*). The two kinase mutants had differing levels of rescuing activity, though both restored apical PAR-6 enrichment in embryos (*Figure 7—figure supplement 1E–G*). The PKC-3[G336N] kinase mutant partially rescued larval lethality and luminal obstruction and was strikingly similar to the CDC-42$^{gut(-)}$ phenotype (*Figure 7C, C', E, F*); by contrast, the PKC-3[K282A] kinase mutant showed almost no larval rescuing activity (*Figure 7D–F*). These results demonstrate that apical PAR-6 localization is not sufficient to restore apical continuity and viability, and that PKC-3 kinase activity and PAR-6-binding are both critical for normal PKC-3 function and for apical continuity.

## PAR-6$^{gut(-)}$ and PKC-3$^{gut(-)}$ intestinal lumens have edematous swellings and multiple constrictions

The intestinal obstructions in PAR-6$^{gut(-)}$ and PKC-3$^{gut(-)}$ larvae could be caused by constricted or closed lumens, so we next examined cell arrangement and tissue morphology with a membrane marker (intestine-specific GFP::CAAX) and DIC imaging. Unlike the continuous open lumen of control intestines (*Figure 8A*), the lumens of PAR-6$^{gut(-)}$ and PKC-3$^{gut(-)}$ intestines frequently appeared to be pinched shut in multiple places, often with edematous swellings (*Figure 8B, C, E*), and PAR-6$^{late\ gut(-)}$ lumens also appeared pinched shut in one or two places in some larvae (*Figure 6—figure supplement 1C*), consistent with a continued requirement for PAR-6 during elongation. CDC-42$^{gut(-)}$ larvae generally had one or two points of luminal constriction (*Figure 8D*), but the overall intestinal morphology was not as severely defective as in PAR-6$^{gut(-)}$ and PKC-3$^{gut(-)}$ larvae. Based on the intercellular position of the midline gaps in embryonic intestines depleted of polarity regulators, we predicted that lumen formation would frequently fail at the interface between adjacent int rings. While control intestines clearly had an open lumen between int rings, PAR-6$^{gut(-)}$ and PKC-3$^{gut(-)}$ intestines all had one or more int interface with a constricted lumen, and CDC-42$^{gut(-)}$ larvae again showed a milder phenotype (*Figure 8F*). Further, most PAR-6$^{gut(-)}$ and PKC-3$^{gut(-)}$ larvae and 50% of CDC-42$^{gut(-)}$ larvae had their lumens pinched shut within int1 (*Figure 8G*), consistent with the int1 obstruction and accumulation of blue-dyed food in the feeding assay and with the anterior star cell midline gaps caused by cell division. Thus, PAR-6 and PKC-3 are essential during embryonic development to build an intestine with a continuous open lumen.

## Apical and junctional proteins are discontinuous in larval PAR-6$^{gut(-)}$ intestines

In examining the membranes of PAR-6$^{gut(-)}$ intestinal cells, the edematous swellings nearly always occurred along the midline, where an apical surface and lumen would normally be built. To further

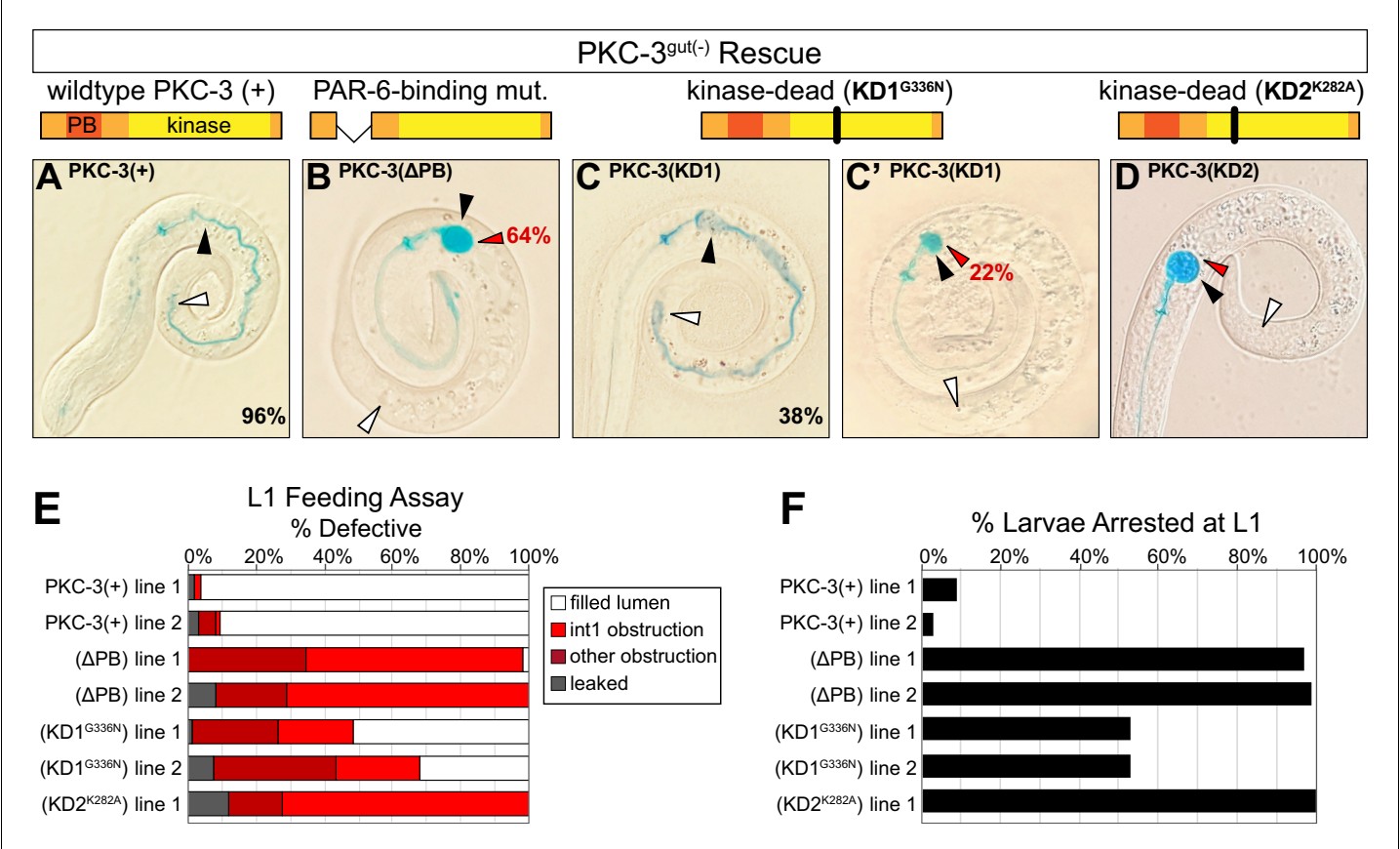

**Figure 7.** PKC-3 kinase and PAR-6-binding activities are essential in the embryonic intestine for larval viability and intestinal function. (A–D) Color images of L1 PKC-3$^{gut(-)}$ larvae carrying a wild-type or mutant PKC-3 transgene expressed in the intestine, with cartoon schematics of PKC-3 mutations shown above the corresponding PKC-3$^{gut(-)}$ larvae, after 3 hr incubation in blue-dyed food. Arrowheads indicate the positions of first intestinal ring int1 (black), the last ring int9 (white), and the first luminal obstruction (red). Percentages indicate frequency of continuous lumen (A, C) and int1 obstruction (B, C', D). (E) Graph showing the average percent of L1 larvae that had defective lumens from three trials of the feeding assay (see *Figure 6—source data 2*). Each transgenic line was independently isolated. (F) Graph showing the average percent of larvae from three trials that arrested in the L1 stage 72 hr after embryos were laid (see *Figure 6—source data 1*). All experiments used *ifb-2*p::*zif-1* to drive E8 onset of PKC-3 degradation and *elt-2p* to drive E4 onset of transgenic intestinal PKC-3 expression.

The online version of this article includes the following figure supplement(s) for figure 7:

**Figure supplement 1.** PAR-6 localization in different PKC-3 backgrounds.

understand how the early gap defects correlated with these later morphological defects, we examined the apical markers PAR-3::GFP and YFP::ACT-5, the junctional marker DLG-1::mNG, and the basolateral marker LGL-1::GFP, which remained expressed in L1 larvae. Unlike the continuous luminal PAR-3 in control L1 larval intestines, PAR-3 localized to hollow sphere-like structures that were often separated by gaps in localization in PAR-6$^{gut(-)}$ larvae (*Figure 9A–C*), consistent with a polarized but discontinuous luminal surface. ACT-5 is strongly expressed in larval intestines and localizes to the apical brush border (*MacQueen et al., 2005*), and unlike the continuous luminal ACT-5 in control larvae, all PAR-6$^{gut(-)}$ larvae and some PAR-6$^{late\ gut(-)}$ larvae showed gaps in luminal ACT-5 (*Figure 9D–F*, *Figure 6—figure supplement 1D*). We also found DLG-1 localization to be defective. While control intestines contained a continuous ladder-like belt arrangement of DLG-1, all PAR-6$^{gut(-)}$ larvae had DLG-1 bands that were clearly discontinuous between adjacent int rings in multiple places (*Figure 9G–I*). Lastly, we examined LGL-1 localization to determine if basolateral proteins continued to mislocalize to the larval intestinal midline. In control larvae, LGL-1 was clearly excluded from apical surfaces along the midline, but in many PAR-6$^{gut(-)}$ larvae, we observed at least partial localization of LGL-1 to midline regions (*Figure 9J–L*), consistent with a continued failure to exclude basolateral

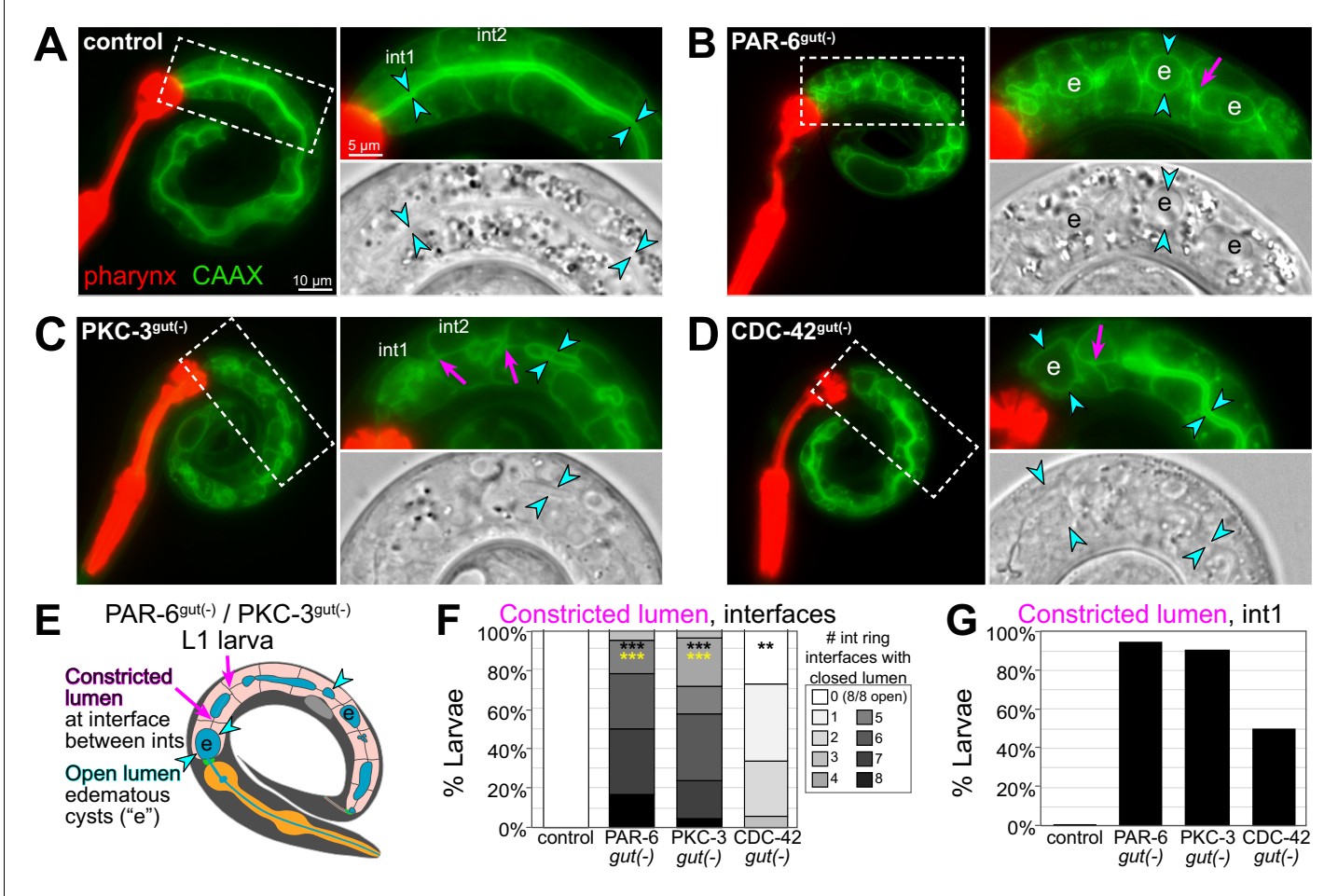

**Figure 8.** PAR-6, PKC-3, and CDC-42 are essential in the embryonic intestine for larval intestine morphology. (A–D) Live images of L1 larvae of indicated genotypes with GFP::CAAX-labeled intestinal membranes and mCherry-labeled pharynges. Maximum intensity Z-projections (0–1 μm) capture the intestinal midline. 2× magnified images of boxed region on the right; GFP::CAAX (top) and DIC micrograph (bottom). Intestinal lumen (paired cyan arrowheads), edematous luminal swellings ('e'), and constricted lumens (magenta arrows) are indicated. (E) Cartoon model of the defects in PAR-6$^{gut(-)}$ and PKC-3$^{gut(-)}$ intestines. (F) Graph showing percent of larvae with the indicated number of int ring interfaces with a constricted lumen (see magenta arrows in E). Control: n = 20, PAR-6$^{gut(-)}$: n = 18, PKC-3$^{gut(-)}$: n = 18, and CDC-42$^{gut(-)}$: n = 21. Statistical analysis: ANOVA with Tukey's post hoc tests. Differences from control indicated with black asterisks. Differences from CDC-42$^{gut(-)}$ indicated with yellow asterisks. No significant difference between PAR-6$^{gut(-)}$ and PKC-3$^{gut(-)}$. (G) Graph showing percent of larvae from (F) with a constricted lumen in int1. All experiments used *ifb-2*p::*zif-1* to drive E8 onset of degradation. Scale bars = 10 μm in (A–D), and 5 μm in insets. **p<0.01, ***p<0.001.

proteins from the midline in PAR-6$^{gut(-)}$ larvae. Together, these results suggest that early gaps in apical and junctional proteins and mislocalized basolateral proteins at the embryonic intestinal midline persist through morphogenesis, leading to larval intestines with discontinuous apical surfaces and junctions separated by midline regions where lumen formation failed to occur.

## Discussion

PAR complex proteins are essential, highly conserved polarity regulators that play critical roles in epithelial cell polarity establishment and for apical and junctional maturation and maintenance. It is often difficult to tease apart their requirement for these different roles, especially in vivo, and to determine how morphological defects caused by their removal during development affect the mature organ. Using tissue-specific protein degradation to deplete proteins from developing intestines, we found that the PAR complex proteins PAR-6, PKC-3, and CDC-42 are required to preserve apical and junctional continuity as cells in the developing intestine divide and elongate during

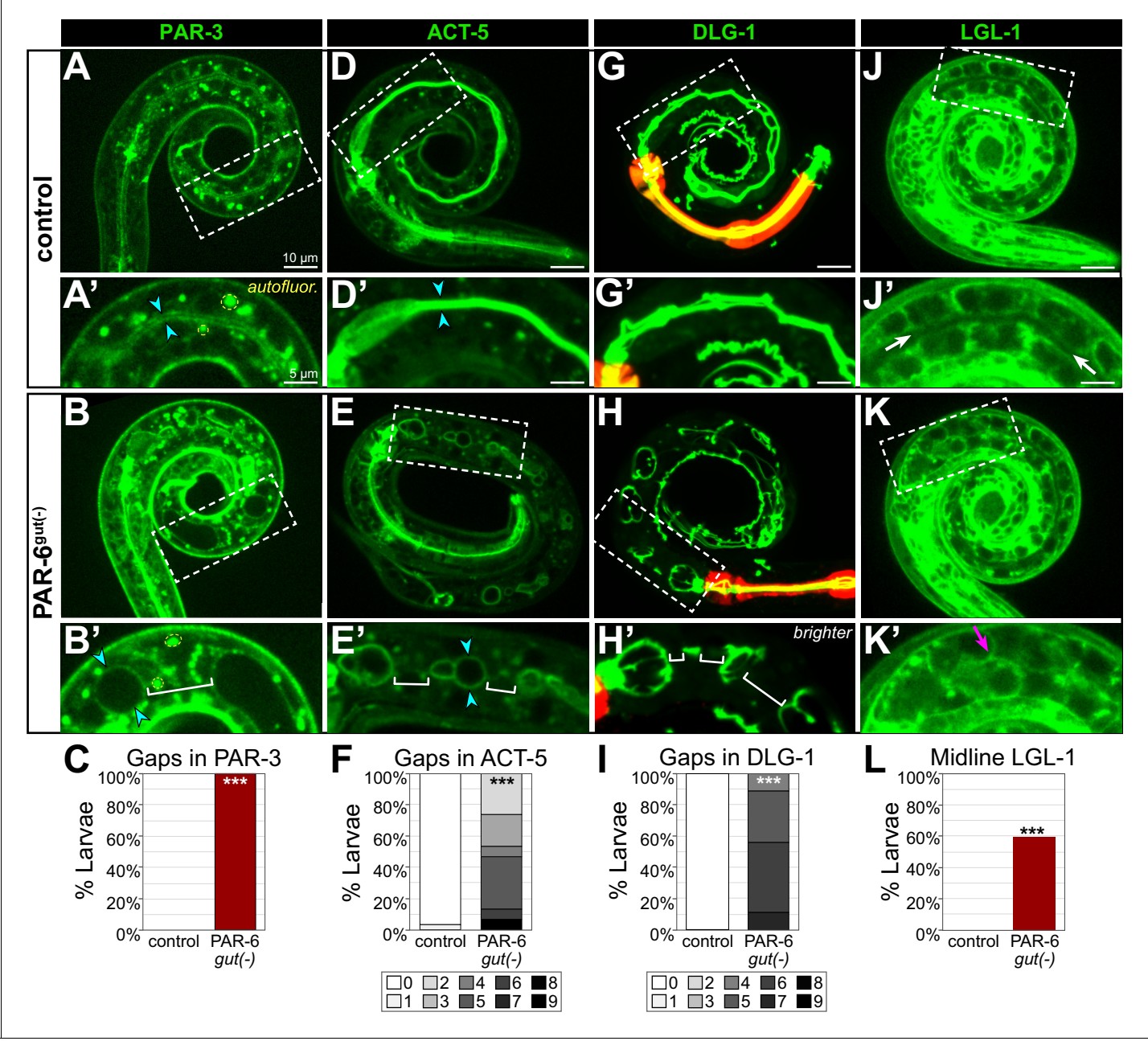

**Figure 9.** PAR-6[gut(-)] larval intestines have gaps in apical and junctional proteins and a discontinuous lumen. (**A-L**) Live images of L1 control and PAR-6[gut(-)] larvae expressing the indicated marker (A, B, D, E, G, H, J, K), with 2× magnified images of boxed region below (A', B', D', E', G', H', J', K'), and graphs quantifying luminal localization defects (C, F, I, L). Maximum intensity Z-projections (0–4 μm) capture the intestinal midline for all images except for the minimum intensity projections (2 μm) of LGL-1::GFP that better visualize the absence of apical LGL-1 in control larvae. Open intestinal lumens (cyan arrowheads), gaps in protein localization (brackets), midline protein localization presence (magenta arrows), or absence (white arrows) are indicated. Yellow dashed circles outline examples of the bright autofluorescent puncta from birefringent gut granules. (**C**) Percent of L1 larvae with any gaps in apical PAR-3::GFP. Control: n = 0/16, PAR-6[gut(-)]: n = 21/21. (**F**) Percent of L1 larvae with indicated number of gaps in apical YFP::ACT-5. Control: n = 29, PAR-6[gut(-)]: n = 15. (**I**) Percent of larvae with indicated number of gaps in DLG-1::mNG. Control: n = 23, PAR-6[gut(-)]: n = 18. (**L**) Percent of L1 larvae with any midline-localized LGL-1::GFP. Control: n = 0/15, PAR-6[gut(-)]: n = 16/27. All experiments used *ifb-2*p::*zif-1* to drive E8 onset of degradation except (**D–F**), which used *elt-2*p::*zif-1* to drive degradation at E4. Scale bars = 10 μm in main panels and 5 μm in insets. Statistical analyses: Student's t-test (**F, I**) and Fisher's exact test (**C, L**). ***p<0.001.

morphogenesis. This continuity is critical for transforming the polarized intestinal primordium into a functional tube at hatching as depletion of any of these proteins resulted in a failure to build a continuous lumen that could pass food. Taken together, our data suggest a model in which PAR-6, PKC-3, and CDC-42 promote apical, junctional, and basolateral domain remodeling to maintain tissue-level continuity during intestinal cell division and elongation, thereby ensuring the formation of an open and functional intestinal lumen (*Figure 10*).

## The contribution of tissue-specific depletion studies

A critical feature of this study was our tissue-specific protein degradation approach. Ubiquitous somatic depletion of PAR-6, PKC-3, or CDC-42 causes embryonic lethality and failed embryonic elongation due to skin defects (*Montoyo-Rosario et al., 2020*; *Totong et al., 2007*; *Zilberman et al., 2017*), thereby masking potential roles of these proteins in other tissues during or after elongation. Using tissue-specific degradation, we bypassed the requirement for these proteins in earlier embryonic development and in other tissues and revealed the requirement for PAR-6, PKC-3, and CDC-42 for apical and junctional continuity during intestinal morphogenesis. Similar techniques have identified different roles and regulation of the PAR complex in the *C. elegans* excretory cell and larval intestine (*Abrams and Nance, 2021*; *Castiglioni et al., 2020*), underscoring the value of this approach in understanding the range of ways that these conserved regulators function. While we found that PAR-6 and PKC-3 are required embryonically to build a functional intestine, they seem to be dispensable once the functional intestine has been built, even though intestinal elongation continues through larval development (*Castiglioni et al., 2020*), suggesting that the embryonic and larval intestines elongate via different mechanisms. These findings thus highlight another advantage of tissue-specific protein depletion: controlling the timing of depletion can uncouple early and late roles in the development and maintenance of the same organ.

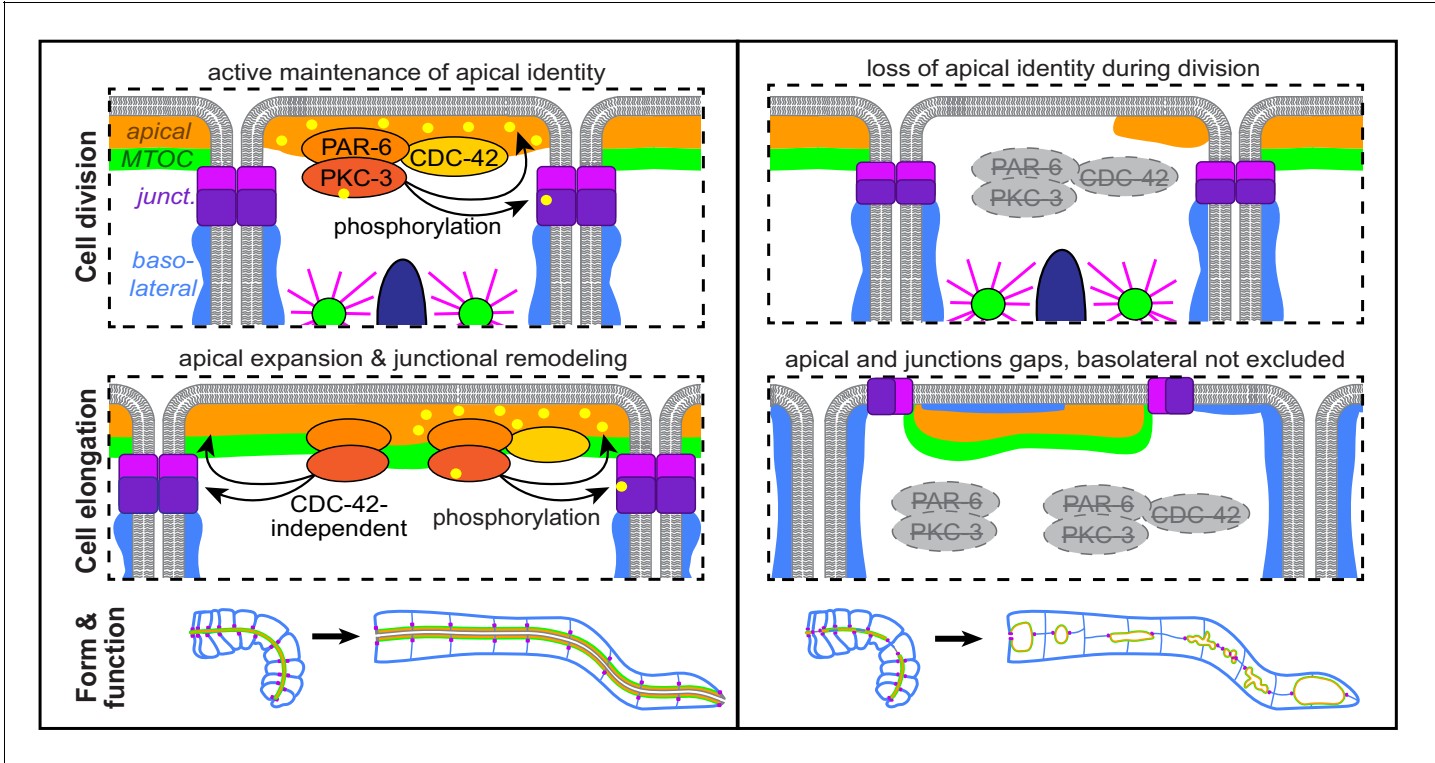

**Figure 10.** Model. A cartoon model of the role of PAR-6, PKC-3, and CDC-42 in apical and junctional remodeling as *C. elegans* embryonic intestinal cells divide and elongate to form a functional intestinal tube.

## PAR complex protein requirement during epithelial cell mitosis

Our study found that dividing cells were sensitive to depletion of PAR-6, PKC-3, and CDC-42 during intestinal morphogenesis. Star cells contribute only three of the nine int rings that build the intestine (*Figure 1—figure supplement 1*), yet over half of the midline gaps in 1.5-fold embryos affected star cell daughters. Cell division appears to cause this high frequency of gap formation as inhibition of the star cell divisions ameliorated the gaps caused by PAR-6 depletion. Dividing epithelial cells extensively remodel many polarized features, any of which could depend on PAR complex proteins. The apical MTOC and microtubules are removed and centrosomes are reactivated to organize microtubules into a bipolar spindle. Apical microtubules contribute to the establishment and maintenance of apical and junctional domains (*Feldman and Priess, 2012*; *Harris and Peifer, 2005*; *Vasileva and Citi, 2018*), so losing apical microtubules during mitosis could weaken the integrity of an epithelial cell and make it more dependent on the PAR complex to maintain polarity. In *Drosophila* neuroblasts, parallel inputs from PAR complex proteins and microtubules assemble a polarized Pins/Gαi crescent to orient the mitotic spindle (*Siegrist and Doe, 2005*). Similarly, apical microtubules and PAR-6 could redundantly maintain the apical domain in star cells; loss of the apical domain would thus only occur when both apical microtubules and PAR-6 are absent, as in dividing PAR-6[gut(-)] star cells.

Division also requires the formation of new junctions between daughter cells (*Baum and Georgiou, 2011*; *Roignot et al., 2013*), a process that could specifically rely on the known ability of PAR complex proteins to regulate junction formation and maturation during polarity establishment. The adherens junctions of interphase cells can promote tissue integrity in dividing neighbor cells by orienting the mitotic spindle, maintaining adhesion, and regulating the length of the new cell-cell interface (*Gloerich et al., 2017*; *Herszterg et al., 2013*; *Higashi et al., 2016*). Synchronous division of neighboring cells therefore creates a new challenge as both cells remodel their junctions at the same time, which could temporarily remove important positional information and adhesion. Indeed, we observed that when wild-type star cells divided synchronously, HMR-1 and DLG-1 levels were drastically reduced. In addition, when PAR-6[gut(-)] star cells divided synchronously, persistent midline gaps in the apical MTOC formed, but we did not observe such gaps during asynchronous divisions. These results suggest that either apical PAR-6 or a non-dividing neighboring cell is sufficient for new daughter cells to correctly remodel and build junctions, but that loss of both can impair this remodeling. Thus, adherens junctions may function as an external orientation cue to ensure correct polarity and junction position in daughter cells.

## Gap formation during epithelial cell elongation: cause versus consequence

Elongating epithelial tubes often must increase the area of the apical domains and junctions of their underlying cells, and consistently, the midline gap phenotype in PAR-6[gut(-)] and PKC-3[gut(-)] embryos became more severe during the greater than threefold increase in intestine length that occurred by the L1 larval stage. In addition, later PAR-6 depletion in elongating intestines also resulted in discontinuous lumens, albeit less frequently, suggesting a continued requirement for PAR-6 during embryonic intestine development. Most gaps occurred at interfaces between anterior and posterior cells without visible physical cell separation, raising the question of what causes cell-cell interfaces to be vulnerable to loss of PAR-6 or PKC-3. One possibility is that weakened junctions allow basolateral domains to invade and compress the apical domain. While we observed some basolateral protein localization in the gaps in PAR-6[gut(-)] embryos, the signal was not specific to the gaps, indicating that basolateral protein localization to the midline does not create apical and junctional gaps. Alternatively, compromised apical and junctional expansion could allow basolateral proteins to diffuse into the new midline space created by elongation. Apical expansion could fail if, for example, the mobility of important apical determinants like PAR-3 decreased and failed to spread to newly added midline membrane. Membrane lacking PAR-3 would likely have less $PI(4,5)P_2$ and no apical microtubules, both of which normally promote apical protein localization and recruitment (*Campanale et al., 2017*; *Rodriguez-Boulan and Macara, 2014*). Failure in junctional remodeling could also impede apical expansion by creating junctions that corral and trap the apical domain as cells elongate. Positioning junctions, apical, and basolateral domains is by necessity a highly

intertwined process as each domain helps reinforce the position of the other, so it is difficult to draw firm conclusions about the causes of the gaps versus the consequences.

## Embryonic PAR complex proteins and intestinal tissue integrity

The primary defect in PAR-6-, PKC-3-, and CDC-42-depleted intestines is a failure to maintain the coordination of apicobasal polarity between cells; apical and junctional gaps correlated with luminal gaps in larval intestines, the luminal gaps blocked the passage of food, and young larvae arrested and died. Despite the lethality, many aspects of epithelial tissue integrity appeared intact in these larval intestines and thus did not require PAR-6, PKC-3, or CDC-42.

### Selective barrier function

Disrupting tight/septate junction proteins can compromise epithelial barrier function and tissue integrity (*Asano et al., 2003*; *Behr et al., 2003*). Intestines depleted of PAR-6, PKC-3, or CDC-42 rarely leaked, so barrier function appeared largely intact. This result was unexpected due to the known requirement for PAR-6 in junction maturation (*Totong et al., 2007*). While junctional DLG-1 localization was dim and fragmented in PAR-6$^{gut(-)}$ embryos (*Totong et al., 2007*, this study), DLG-1 appeared to recover a junction-like localization in PAR-6$^{gut(-)}$ larvae. In addition, DLG-1 was usually visibly continuous between the anterior valve and int1, which may explain why their barrier function remained intact. This result suggests that a secondary mechanism may drive later septate-like junction formation independent of PAR-6 and PKC-3. A candidate parallel pathway is the transmembrane Crumbs polarity complex, a critical polarity regulator in many epithelia (*Charrier et al., 2015*; *Tepass et al., 1990*). Simultaneous removal of all three *C. elegans* Crumbs homologs does not cause obvious polarity defects (*Waaijers et al., 2015*), but their role could normally be masked by the activity of the PAR complex.

### Lumen formation

Lumenogenesis requires correct apicobasal polarity as apically directed vesicles add membrane, apical proteins, and channel and pump proteins to expand the apical surface and build the luminal space (*Datta et al., 2011*; *Shafaq-Zadah et al., 2020*). Yet without PAR-6, larval intestines developed luminal regions surrounded by apical proteins, suggesting that the ability of the apical domain to direct apical trafficking and lumenogenesis was intact. The edematous nature of much of the PAR-6$^{gut(-)}$ and PKC-3$^{gut(-)}$ lumens may indicate an inability to regulate luminal expansion. However, luminal expansion is usually associated with increased apical polarity proteins, such as overexpressed Crumbs, or decreased basolateral proteins (*Datta et al., 2011*), the opposite of our depletion cases. Therefore, we favor a model in which the edematous swellings are the by-product of normal lumenogenesis at the apical surface but with no exit route for the luminal fluid, leading to swelling due to trapped fluid accumulation.

An important caveat to our interpretations is that tissue-specific degradation is not equivalent to a null allele. While we see robust degradation of all three PAR complex proteins (*Figure 2—figure supplement 1*, *Figure 7—figure supplement 1*), we cannot exclude the possibility that low levels of undegraded protein are sufficient to rescue lumenogenesis and barrier function. However, the level of PAR-6 and PKC-3 degradation we achieve is clearly sufficient to cause fully penetrant luminal gaps and larval arrest. Thus, we consider the likeliest explanation to be that PAR-6, PKC-3, and CDC-42 are not required in the developing intestine to build a lumen per se or to maintain barrier function. Rather, these proteins maintain apical and junctional domain continuity and thus continuous lumenogenesis, resulting in a functional intestinal tube.

## PAR-6 and PKC-3 play a more substantial role than CDC-42 in maintaining apical continuity

In many tissues, CDC-42 is required for PAR complex localization and to activate PKC-3 kinase activity (*Abrams and Nance, 2021*; *Campanale et al., 2017*; *Pichaud et al., 2019*), but CDC-42 is not required to establish apical polarity or to recruit apical PAR-6 in the intestine (*Zilberman et al., 2017*). Our results revealed two surprising findings about CDC-42. First, by depleting CDC-42 specifically in the intestine, we bypassed the embryonic lethality caused by skin-based enclosure defects (*Zilberman et al., 2017*), and we were able to discover a later role for CDC-42 in promoting

intestinal apical continuity after apical polarity is established. Second, while CDC-42$^{gut(-)}$, PAR-6$^{gut(-)}$, and PKC-3$^{gut(-)}$ embryos all had similar embryonic gap defects, we found that the number of intestinal gaps defects increased only in PAR-6$^{gut(-)}$ and PKC-3$^{gut(-)}$ larvae, and not in CDC-42$^{gut(-)}$ larvae. This result indicates that PAR-6 and PKC-3 continue to be required to maintain apical continuity as the embryonic intestine further elongates, whereas CDC-42 may only be required early in intestinal morphogenesis. One possible explanation for the difference in larval phenotypic severity could be that CDC-42 is important for activating all PKC-3 kinase activity, but that PAR-6 and PKC-3 also provide kinase-independent activity, as has been observed in the *Drosophila* embryonic ectoderm (*Kim et al., 2009*). Consistent with this possibility, the PKC-3 kinase mutant G336N partially rescued PKC-3$^{gut(-)}$, resulting in a phenotype similar to CDC-42$^{gut(-)}$ larvae. Alternatively, G336N could be acting as a hypomorphic mutation as a second kinase mutant K282A completely abolished PKC-3 rescuing activity.

Apicobasal polarity regulators are highly conserved, yet differentially required across organisms and tissues (*Pickett et al., 2019*; *Wen and Zhang, 2018*). Similarly, the same gene can have oncogenic or tumor-suppressive functions in different tissues; for example, *PARD3*/Par3 overexpression is associated with renal cancers, but *PARD3* downregulation or deletion is associated with breast, glioblastoma, lung, and other cancers (*Halaoui and McCaffrey, 2015*). Detailed studies of diverse epithelia will continue to be important for parsing apart the roles of critical polarity proteins during different stages of organ development and homeostasis, to deepen our understanding about the many ways in which these highly conserved regulators function in development and in disease.

# Materials and methods

## Key resources table

| Reagent type (species) or resource | Designation | Source or reference | Identifiers | Additional information |
| --- | --- | --- | --- | --- |
| Strain, strain background (*Caenorhabditis elegans*) | JLF631 | This paper; *Sallee et al., 2018*; *Asan et al., 2016*; *Wang et al., 2015* | | *ltSi569*[TBG-1::mCherry]; *zif-1(gk117)*; *wowIs3*[*ifb-2*p::zif-1, *myo-2*p::mCherry] *zuIs70*[*end-1*p::GFP::CAAX] |
| Strain, strain background (*C. elegans*) | JLF37 | This paper; *Sallee et al., 2018* | | *zuIs278*[*pie-1*p::mCherry::TBA-1]; *gip-1*(*wow5*[ZF::GFP::GIP-1]) *zif-1(gk117)* |
| Strain, strain background (*C. elegans*) | JLF83 | This paper; *Sallee et al., 2018* | | *ltSi569*; *ptrn-1*(*wow4*[PTRN-1::GFP]) |
| Strain, strain background (*C. elegans*) | JLF153 | This paper; *Sallee et al., 2018* | | *ltSi569*; *zif-1(gk117)*; *zyg-9*(*wow13*[ZYG-9::ZF::GFP]) |
| Strain, strain background (*C. elegans*) | JJ2376 | CGC, Asako Sugimoto | | *ddIs6*[*pie-1*p::GFP::TBG-1]; *tjIs222*[*pie-1*p::mCherry::AIR-1] |
| Strain, strain background (*C. elegans*) | JLF152 | This paper; *Sallee et al., 2018* | | *ltSi569*; *zif-1(gk117)*; *noca-1*(*wow11*[NOCA-1::ZF::GFP]) |
| Strain, strain background (*C. elegans*) | JLF729 | This paper; *Sanchez et al., 2020* | | *ltSi569 vab-10*(*wow 80*[VAB-10B::ZF::GFP]); *zif-1(gk117)* |
| Strain, strain background (*C. elegans*) | JLF878 | This paper; *Heppert et al., 2018* | | *ltSi569*; *zif-1(gk117)*; *wowIs3*; *dlg-1*(*cp 301*[DLG-1::mNeon Green]) |
| Strain, strain background (*C. elegans*) | JLF719 | This paper; *Marston et al., 2016* | | *hmr-1*(*cp21*[HMR-1::GFP]) *ltSi569*; *zif-1(gk117)*; *wowIs3* |

*Continued on next page*

*Continued*

| Reagent type (species) or resource | Designation | Source or reference | Identifiers | Additional information |
|---|---|---|---|---|
| Strain, strain background (*C. elegans*) | JLF442 | This paper; *Neukomm et al., 2011* | | *zuIs278; opIs310[ced-1p::YFP::ACT-5]* |
| Strain, strain background (*C. elegans*) | JLF148 | This paper; CGC | | *ltSi569 par-6(it319[PAR-6::GFP]); unc-119(ed3 or +)* |
| Strain, strain background (*C. elegans*) | JLF149 | This paper; CGC | | *ltSi569; pkc-3(it309[GFP::PKC-3]); unc-119(ed3) or unc-119(+)* |
| Strain, strain background (*C. elegans*) | JLF147 | This paper; CGC | | *ltSi569; par-3(it298[PAR-3::GFP]); unc-119(ed3) or unc-119(+)* |
| Strain, strain background (*C. elegans*) | JLF440 | This paper | | *ltSi569; zif-1(gk117); wowIs3; ptrn-1(wow4)* |
| Strain, strain background (*C. elegans*) | JLF445 | This paper | | *ltSi569 par-6(wow31[PAR-6::ZF::GFP])/hT2[qIs48]; zif-1(gk117)/hT2; wowIs3; ptrn-1(wow4)* |
| Strain, strain background (*C. elegans*) | JLF492 | This paper | | *ltSi569; pkc-3(wow85[ZF::GFP::PKC-3])/mIn1[mIs14]; zif-1(gk117); wowIs3; ptrn-1(wow4)* |
| Strain, strain background (*C. elegans*) | JLF632 | This paper | | *ltSi569 par-6(wow31)/hT2[qIs48]; zif-1(gk117)/hT2; wowIs3 zuIs70* |
| Strain, strain background (*C. elegans*) | JLF876 | This paper | | *ltSi569; pkc-3(wow85)/mIn1[mIs14]; zif-1(gk117); wowIs3 zuIs70* |
| Strain, strain background (*C. elegans*) | JLF877 | This paper; *Zilberman et al., 2017* | | *ltSi569; cdc-42(xn65[ZF::YFP::CDC-42])/mIn1[mIs14]; zif-1(gk117); wowIs3 zuIs70* |
| Strain, strain background (*C. elegans*) | JLF1058 | This paper; *Lee et al., 2016* | | *ltSi569; cdc-25.2(ok597)/oxTi980; zif-1(gk117); wowIs3 zuIs70* |
| Strain, strain background (*C. elegans*) | JLF1020 | This paper | | *ltSi569 par-6(wow31)/hT2[qIs48]; cdc-25.2(ok597)/oxTi980; zif-1(gk117)/hT2; wowIs3 zuIs70* |
| Strain, strain background (*C. elegans*) | JLF724 | This paper; CGC | | *ltSi569 par-6(wow31)/tmC27[unc-75(tmIs1239[myo-2p::Venus])]; wowIs28[elt-2p::zif-1, end-1p::histone::mCherry, myo-2p::mCherry; zif-1(gk117)]; opIs310* |
| Strain, strain background (*C. elegans*) | JLF784 | This paper | | *par-3(wow121[PAR-3::tagRFP]) zif-1(gk117); wowIs3; wow4* |
| Strain, strain background (*C. elegans*) | JLF785 | This paper | | *par-6(wow119[PAR-6::ZF::GFP11])/hT2[qIs48]; zif-1(gk117)/hT2; wowIs3; wow4* |
| Strain, strain background (*C. elegans*) | JLF720 | This paper | | *hmr-1(cp21) ltSi569 par-6(wow31)/hT2[qIs48]; zif-1(gk117)/hT2; wowIs3* |

*Continued on next page*

*Continued*

| Reagent type (species) or resource | Designation | Source or reference | Identifiers | Additional information |
|---|---|---|---|---|
| Strain, strain background (*C. elegans*) | JLF721 | This paper | | *ltSi569 par-6(wow31)/ hT2[qIs48]; zif-1(gk117)/ hT2; wowIs3; dlg-1(cp301)* |
| Strain, strain background (*C. elegans*) | JLF880 | This paper; **Beatty et al., 2010** | | *ltSi569; itIs256[LGL-1::GFP]; zif-1(gk117); wowIs3* |
| Strain, strain background (*C. elegans*) | JLF881 | This paper | | *ltSi569 par-6(wow31)/ hT2[qIs48]; itIs256; zif-1(gk117)/hT2; wowIs3* |
| Strain, strain background (*C. elegans*) | JLF879 | This paper; **Legouis et al., 2000** | | LET-413::GFP *par-3(wow121[PAR-3::tagRFP]) zif-1(gk117); wowIs3* |
| Strain, strain background (*C. elegans*) | JLF817 | This paper | | *par-6(wow119)/hT2[qIs48];* LET-413::GFP *par-3(wow121) zif-1(gk117)/hT2; wowIs3* |
| Strain, strain background (*C. elegans*) | JLF204 | This paper | | *ltSi569; zif-1(gk117); wowIs3* |
| Strain, strain background (*C. elegans*) | JLF1056 | This paper | | *zif-1(gk117); wowEx175[asp-1p::zif-1; myo-2p::mCherry]* |
| Strain, strain background (*C. elegans*) | JLF212 | This paper | | *par-6(wow31); zif-1(gk117)* |
| Strain, strain background (*C. elegans*) | JLF758 | This paper | | *par-6(wow119); zif-1(gk117)* |
| Strain, strain background (*C. elegans*) | JLF480 | This paper | | *pkc-3(wow85); zif-1(gk117)* |
| Strain, strain background (*C. elegans*) | JLF882 | This paper | | *cdc-42(xn65); zif-1(gk117)* |
| Strain, strain background (*C. elegans*) | JLF354 | This paper | | *ltSi569 par-6(wow31)/ hT2[qIs48]; zif-1(gk117)/ hT2; wowIs3* |
| Strain, strain background (*C. elegans*) | JLF726 | This paper | | *par-6(wow119)/hT2[qIs48]; zif-1(gk117)/hT2; wowIs3* |
| Strain, strain background (*C. elegans*) | JLF491 | This paper | | *pkc-3(wow85)/mIn1[mIs14]; zif-1(gk117); wowIs3* |
| Strain, strain background (*C. elegans*) | JLF883 | This paper | | *ltSi569; cdc-42(xn65)/ mIn1[mIs14]; zif-1(gk117); wowIs3* |
| Strain, strain background (*C. elegans*) | JLF1072 | This paper | | *par-6(wow31)/hT2; zif-1(gk117)/hT2[qIs48]; wowEx175* |
| Strain, strain background (*C. elegans*) | JLF1073 | This paper | | *ltSi569 par-6(wow31)/hT2; zif-1(gk117)/hT2[qIs48]; zuIs70; wowEx175* |
| Strain, strain background (*C. elegans*) | JLF866 | This paper | | *pkc-3(wow85); zif-1(gk117); wowIs3; wowEx143[elt-2p::BFP::PKC-3(+); unc-122p::gfp]* 'line 1' |

*Continued on next page*

Continued

| Reagent type (species) or resource | Designation | Source or reference | Identifiers | Additional information |
|---|---|---|---|---|
| Strain, strain background (*C. elegans*) | JLF867 | This paper | | *pkc-3(wow85); zif-1(gk117); wowIs3; wowEx144[elt-2p::BFP::PKC-3(+); unc-122p::gfp]* 'line 2' |
| Strain, strain background (*C. elegans*) | JLF870 | This paper | | *pkc-3(wow85)/mIn1[mIs14]; zif-1(gk117); wowIs3; wowEx150[elt-2p::BFP::PKC-3(ΔPB); unc-122p::gfp]* 'line 1' |
| Strain, strain background (*C. elegans*) | JLF1044 | This paper | | *pkc-3(wow85)/mIn1[mIs14]; zif-1(gk117); wowIs3; wowEx148[elt-2p::BFP::PKC-3(ΔPB); unc-122p::gfp]* 'line 2' |
| Strain, strain background (*C. elegans*) | JLF868 | This paper | | *pkc-3(wow85)/mIn1[mIs14]; zif-1(gk117); wowIs3; wowEx146[elt-2p::BFP::PKC-3(G336N); unc-122p::gfp]* 'line 1' |
| Strain, strain background (*C. elegans*) | JLF869 | This paper | | *pkc-3(wow85)/mIn1[mIs14]; zif-1(gk117); wowIs3; wowEx147[elt-2p::BFP::PKC-3(G336N); unc-122p::gfp]* 'line 2' |
| Strain, strain background (*C. elegans*) | JLF1074 | This paper | | *pkc-3(wow85)/mIn1[mIs14]; zif-1(gk117); wowIs3; wowEx184[elt-2p::BFP::PKC-3(K282A); unc-122p::gfp]* 'line 1' |
| Strain, strain background (*C. elegans*) | JLF895 | This paper | | *ltSi569; par-3(wow 120[PAR-3::GFP]) zif-1(gk117); wowIs3* |
| Strain, strain background (*C. elegans*) | JLF884 | This paper | | *ltSi569 par-6(wow31)/ hT2[qIs48]; par-3(wow120) zif-1(gk117)/hT2; wowIs3* |
| Strain, strain background (*C. elegans*) | JLF885 | This paper | | *ltSi569; pkc-3(wow85)/ mIn1[mIs14]; zif-1(gk117); wowIs3* |
| Strain, strain background (*C. elegans*) | JLF877 | This paper | | *ltSi569; cdc-42(xn65)/ mIn1[mIs14]; zif-1(gk117); wowIs3* |
| Strain, strain background (*C. elegans*) | JLF1075 | This paper | | *zif-1(gk117); zuIs70; wowEx175* |
| Strain, strain background (*C. elegans*) | JLF1088 | This paper | | *ltSi569 par-6(wow31)/ hT2[qIs48]; zif-1(gk117)/ hT2; opIs310; wowEx175* |
| Strain, strain background (*C. elegans*) | JLF1076 | This paper | | *par-6(wow159[PAR-6::RFP]); pkc-3(wow85); zif-1(gk117)* |
| Strain, strain background (*C. elegans*) | JLF1077 | This paper | | *par-6(wow159); pkc-3(wow85)/mIn1[mIs14]; zif-1(gk117); wowIs3; wowEx143[PKC-3(+)]* |
| Strain, strain background (*C. elegans*) | JLF1078 | This paper | | *par-6(wow159); pkc-3(wow85)/mIn1[mIs14]; zif-1(gk117); wowIs3; wowEx148[ΔPB]* |

*Continued on next page*

*Continued*

| Reagent type (species) or resource | Designation | Source or reference | Identifiers | Additional information |
|---|---|---|---|---|
| Strain, strain background (*C. elegans*) | JLF1079 | This paper | | *par-6(wow159); pkc-3(wow85)/ mln1[mls14]; zif-1(gk117); wowIs3; wowEx146[PKC-3(G336N)]* |
| Strain, strain background (*C. elegans*) | JLF1080 | This paper | | *par-6(wow159); pkc-3(wow85)/ mln1[mls14]; zif-1(gk117); wowIs3; wowEx184[PKC-3(K282A)]* |
| Recombinant DNA reagent | Plasmid: pDD162 | Addgene | | Cas9 + sgRNA plasmid template |
| Recombinant DNA reagent | Plasmid: pJF250 | *Sallee et al., 2018* | | ZF::GFP SEC plasmid template |
| Recombinant DNA reagent | Plasmid: pLC01 | Lauren Cote | | ZF::GFP(11) SEC plasmid template |
| Recombinant DNA reagent | Plasmid: pAL29 | Alex Lessenger, James McGhee | | *asp-1p::zif-1* gDNA |
| Recombinant DNA reagent | Plasmid: pCFJ90 | Addgene | | *myo-2*p::*mCherry* |
| Recombinant DNA reagent | Plasmid: pMS252 | This paper | | *elt-2*p::*bfp::pkc-3*(+) gDNA |
| Sequence-based reagent | oligo: oMS-201-F | This paper | AGCTCGGACA CAAGCTCAA CagcgctTCG TCTCCGA CATCATTAGAGGAG | pkc-3gDNA Fwd with BFP overlap |
| Sequence-based reagent | oligo: oMS202-R | This paper | atgttgaagag taattggac TCAGACTGAATCTTC CCGACTCATTTG | pkc-3gDNA Rev with unc-54 overlap |
| Recombinant DNA reagent | Plasmid: pMS259 | This paper | | *elt-2*p::*bfp::pkc-3[G336N]* gDNA |
| Sequence-based reagent | oligo: oMS-235-F | This paper | ATTCGTTCCTaat GGTGATCTGATG | pkc-3(G336N) Fwd |
| Sequence-based reagent | oligo: oMS-236-R | This paper | TCGATGACA AAGAACAGG | pkc-3(G336N) Rev |
| Recombinant DNA reagent | Plasmid: pMS264 | This paper | | *elt-2*p::*bfp::pkc-3[K282A]* gDNA |
| Sequence-based reagent | oligo: oMP215 | This paper | CGCGATAgcA ATTATCAAAAA | PKC-3(K282A) Fwd |
| Sequence-based reagent | oligo: oMP216 | This paper | TAAATTTGACGAG TTGAAACATG | PKC-3(K282A) Rev |
| Recombinant DNA reagent | Plasmid: pMS260 | This paper | | *elt-2*p::*bfp::pkc-3[deltaPB]* gDNA |
| Sequence-based reagent | oligo: oMP213 | This paper | AAACCAGAGC TGCCCGGG | PKC-3(delta PB) Fwd |
| Sequence-based reagent | oligo: oMP214 | This paper | AGCGCTGTTGA GCTTGTGTC | PKC-3(delta PB) Rev |
| Other | | | | CRISPR Allele DNA reagents: see *Supplementary file 1* |

## *C. elegans* strains and maintenance

Nematodes were maintained at 20°C and cultured and manipulated as previously described (*Sulston and Brenner, 1974*), unless otherwise stated. Experiments were performed using embryos

and larvae collected from 1- or 2-day-old adults. The strains used in this study are listed in the Key resources table.

## New alleles and transgenes
### CRISPR cloning and editing
The self-excision cassette (SEC) method was used to generate CRISPR alleles (*Dickinson et al., 2015*). GFP or ZF::GFP alleles also contain 3× FLAG and TagRFP-T alleles also contain 3× Myc. The plasmid pDD162 used to deliver Cas9 and each sgRNA was modified (Q5 Site-Directed Mutagenesis Kit, NEB) to insert the appropriate sgRNA guide sequence for each CRISPR edit. The repair template was generated by PCR-amplifying appropriate homology arm sequences for an N-terminal, C-terminal, or internal fluorophore insertions (Phusion High-Fidelity DNA polymerase, Thermo Scientific) and cloned into an SEC backbone plasmid (NEBuilder HiFi DNA Assembly Master Mix, NEB). The modified Cas9/sgRNA plasmid, repair template, and pBS were injected at 50 ng/µL each into N2 or *zif-1(gk117)* mutant 1-day-old adult worms. Injected worms were recovered and treated according to published protocols to isolate independent CRISPR edit events and excise the SEC. One exception is the PAR-6::tagRFP allele, for which the SEC was not excised. New CRISPR alleles were backcrossed at least twice before being used for subsequent experiments. The internal in-frame placement of the fluorophore in PAR-3 tags all isoforms without causing an obvious phenotype. PAR-6 was tagged at the C-terminus with ZF::GFP and ZF::GFP$_{11}$. Both ZF-tagged *par-6* alleles resulted in fully penetrant larval arrest when degraded in the intestine (*Figure 6A*). sgRNA and homology arm sequences, plasmids, and primers used in constructing new CRISPR alleles are listed in *Supplementary file 1*.

Integrated and extrachromosomal arrays *wowIs3[ifb-2p::zif-1]* (IV or V) was derived by spontaneous integration of the extrachromosomal array *wowEx34* (*Sallee et al., 2018*). *wowIs28[elt-2p::zif-1]* (II) was derived by spontaneous integration of an extrachromosomal array carrying SA109 (*elt-2p::zif-1* at 50 ng/µL, *Armenti et al., 2014*), pJF248 (*end-1p::histone::mCherry* at 50 ng/µL), pCFJ90 (*myo-2p::mCherry* at 2.5 ng/µL, *Frøkjaer-Jensen et al., 2008*), and pBS (47.5 ng/µL). The YFP::ACT-5 transgene *opIs310* was genetically linked to *wowIs3*, so *wowIs28[elt-2p::zif-1]* (II) was used instead for intestine-specific ZIF-1 expression.

An extrachromosomal array driving late degradation of PAR-6 in the embryonic intestine was generated by injecting pAL29 (*asp-1p::zif-1* at 50 ng/µL), pCFJ90 (2.5 ng/µL), and pBS (47.5 ng/µL) into young adult *zif-1(0)* hermaphrodites. The 3 kb *asp-1* promoter for pAL29 came from pJM481, a gift from James McGhee. Extrachromosomal arrays were also generated to provide intestine-specific expression of wild-type or mutant PKC-3. The plasmid pMS252 containing *elt-2p::bfp::pkc-3::unc-54* 3′UTR was generated by PCR-amplifying *pkc-3* from N2 wild-type genomic DNA and the *unc-54* 3′UTR sequence from pSA109 with Phusion DNA polymerase, and using the NEBuilder Assembly Mix, PCR products were cloned into a plasmid carrying *elt-2p::bfp* (pMP27) digested with AfeI and XmaI. Mutations were introduced by Q5 mutagenesis to make pMS259 carrying a PKC-3(G336N) mutation (forward primer: ATTCGTTCCTaatGGTGATCTGATG; reverse primer: TCGATGACAAA-GAACAGG), pMS264 carrying a PKC-3(K282A) mutation (forward primer: CGCGATAGcAATTA TCAAAAA; reverse primer: TAAATTTGACGAGTTGAAACATG), and pMS260 carrying a deletion of the PKC-3 N-terminus carrying the PB1 domain (forward primer: AAACCAGAGCTGCCCGGG; reverse primer: AGCGCTGTTGAGCTTGTGTC). For the PKC-3 rescue assay, *elt-2p::bfp::pkc-3* gDNA plasmids were each injected at 10 ng/µL into JLF148 or JLF155 young adult hermaphrodites along with pBS (115 ng/µL) as carrier DNA and *unc-122p::GFP* (25 ng/µL) to mark the coelomocytes, and independent lines were isolated and crossed into JLF491 to test for function.

## Obtaining gut(-) depletion embryos and larvae
All PAR-6, PKC-3, and CDC-42 intestine-specific depletion strains were maintained with the ZF::GFP allele balanced by the pharyngeal GFP-marked *hT2* (PAR-6), *tmC27* (PAR-6), or *mIn1* (PKC-3, CDC-42). To obtain 'gut(-)' embryos, hermaphrodite L3 and L4 larvae lacking the balancer were transferred to a fresh plate, and their embryos were scored for defects so that both maternal and zygotic supplies of each protein was ZF::GFP-tagged and thus susceptible to degradation. Depletion was verified by examining the loss of GFP fluorescence at the intestinal midline (*Figure 2—figure supplement 1*, *Figure 6—figure supplement 1A, B*, *Figure 7—figure supplement 1H*). While this

strategy does not generate a null situation, this system robustly depletes ZF-tagged proteins (*Liang et al., 2020*; *Sallee et al., 2018*; *Sanchez et al., 2020*; *Magescas et al., 2021*), and we see strong degradation with only occasional and extremely weak midline GFP signal being detectable. This occasional weak GFP signal was only observed when the *wowIs3* transgene began to be silenced over time in a few strains and could be identified by the visible loss of pharyngeal mCherry expression, a co-injection marker of the transgene. Also, rare 'escaper' larvae did not arrest at the L1 stage when silencing occurred.

## Microscopy

Embryos were raised at 20°C and either dissected from gravid hermaphrodites after a 5 hr incubation in M9 for time courses and star cell divisions, or collected from plates containing 1-day-old gravid adults. Embryos were mounted on a pad made of 3% agarose dissolved in M9 and imaged using a Nikon Ti-E inverted microscope (Nikon Instruments, Melville, NY) with a 60× Oil Plan Apochromat (NA = 1.4) objective controlled by NIS Elements software (Nikon). Images were acquired with an Andor Ixon Ultra back thinned EM-CCD camera using 488 nm or 561 nm imaging lasers and a Yokogawa X1 confocal spinning disk head equipped with a 1.5× magnifying lens. Images were taken at a z-sampling rate of 0.5 μm. Live time-lapse imaging was done with 4 min time steps for star cell divisions (*Figure 2*) and with 10 min steps for elongation and *asp-1*p::*zif-1* degradation timing (*Figure 3*, *Figure 6—figure supplement 1A, B*).

For fluorescence imaging in L1 larvae, young adult hermaphrodites were allowed to lay eggs overnight, and larvae were picked into a drop of 2 mM levamisole on a 3% agarose pad to minimize movement, and imaged on a confocal spinning disk microscope (described above), or a Nikon Ni-E compound microscope with an Andor Zyla sCMOS camera with a 60× Oil Plan Apochromat (NA = 1.4) objective and NIS Elements software. For imaging larvae for the 'Smurf' feeding assay, a Vankey Cellphone Telescope Adapter Mount (Amazon) was used with an Apple iPhone 7 or 12mini camera and the compound microscope 60× objective. Brightness, contrast, and hue were adjusted for each color image with Adobe Photoshop v21.0.1.

Percent laser power and exposure time were the same for all genotypes imaged within each experiment. Images were processed in NIS Elements and the Fiji distribution of ImageJ ('Fiji,' *Schindelin et al., 2012*).

## Image quantification

### General parameters for fluorescent signal quantification

Fiji software was used to measure fluorescent signal intensities in dorsal or lateral embryos of the indicated stages. Prior to enrichment and intensity calculations, slide background (average signal intensity of three large ROIs far from embryo) was measured for each image and subtracted from apical, junctional, lateral, and/or cytoplasmic intensity measurements. Signal intensity at apical, junctional, or lateral surfaces was measured by drawing a two pixel-wide segmented line (or four pixel-wide for *Figure 3* TBG-1; PTRN-1 analysis) along the indicated surface, carefully avoiding pharyngeal and rectal valve signal or germ cell signal from outside the intestine, and determining the average signal intensity. Cytoplasmic signal intensity was measured by averaging the signal intensity along three short lines (or a single line >10 μm for *Figure 3* TBG-1; PTRN-1 analysis), taking care to avoid the midline, germ cells, and nuclei as best as possible.

Enrichment = (average apical, junctional, or lateral intensity – average intestinal cytoplasmic intensity)/(average intestinal cytoplasmic intensity).

Signal intensity = (average apical, junctional, or lateral intensity – average intestinal cytoplasmic intensity).

### Specifics for MTOC gap analysis with TBG-1; PTRN-1 (*Figure 3*)

Midline signal fluorescence intensity of late comma- to 1.8-fold-stage embryos was measured on a maximum projection of the z-slices that capture the entire intestinal midline signal. Each measurement was normalized as follows: (measurement – average intestinal background intensity)/(average intestinal background intensity). We reasoned that midline signal intensity in a gap should be in the same range as intestinal cytoplasmic background signal intensity, so we used intestinal background variability to define a 'background-level' measurement. 97.5% of normalized intestinal background

measurements were less than 2 standard deviations above 0, so we used this cutoff (0 + 2 standard deviations) to distinguish between a 'background-level' and apical enrichment-level measurement at the midline. A normalized midline measurement that was background level for both TBG-1 and PTRN-1 was considered a background-level measurement in our analysis. We defined a 'gap' in the midline as a region of the midline with at least three consecutive background-level measurements. This is likely a conservative measurement of gaps as some gaps were visible to the eye that were not detected by this analysis (compare *Figure 3F* to *Figure 4E*). We did not distinguish between star cell and non-star cell gaps in this analysis.

For time-lapse imaging of elongation (*Figure 3H*), we performed the above analysis at two time points. The first time point was the first image frame after the anterior cell divisions had completed, and the second time point was 50 min later at approximately 1.5-fold stage.

### Specifics for apical and lateral signal intensity analysis (*Figure 5*)

Midline ('mid') signal fluorescence intensity of late comma- to 1.8-fold-stage embryos was measured on a single z-slice that captured anterior intestinal midline signal in control embryos, along with the average signal fluorescence along three lateral surfaces ('lat'). In addition, in PAR-6<sup>gut(-)</sup> embryos, a line was drawn along the midline in the visible apical gap ('gap') as well as in the neighboring cell(s) in which the apical marker was localized ('mid'). For *Figure 5*, each plot point represents the measurement of the midline or lateral surfaces of a single embryo in controls, and in PAR-6<sup>gut(-)</sup> backgrounds, each plot point represents a single gap and associated midline and lateral measurement, with a maximum of two measurements per embryo.

### Specifics for apical and junctional protein enrichment (*Figure 1—figure supplement 2*)

Apical and junctional signal intensity was measured on a maximum projection of the z-slices that captured the intestinal midline signal of the two anterior star cells and their two posterior non-star cell neighbors. A 'Pre-division' embryo was defined as a bean stage embryo with a polarized pharynx in which the star cell divisions had not yet occurred. 'During division' embryos were defined by the synchronous star cell divisions and visible apical gap in TBG-1::mCherry. For apical proteins, lines were drawn along the midline, and for junctional proteins, lines were drawn along the junctions and averaged.

### Specifics for apical PAR-6 and PKC-3 enrichment (*Figure 7—figure supplement 1*)

Apical signal intensity of PAR-6::tagRFP and ZF::GFP::PKC-3 in late comma- to 1.8-fold-stage embryos in *Figure 7—figure supplement 1G, H* was measured by drawing a line along the intestinal midline on a maximum projection of the z-slices that captured the intestinal midline signal.

### Blocked star cell divisions in *cdc-25.2(0)* embryos

To prevent cell division in polarized PAR-6<sup>gut(-)</sup> intestines, we used the *cdc-25.2(ok597)* null allele to block the intestinal star cell divisions (*Lee et al., 2016*). We found that occasionally some of the earlier intestinal cell divisions were blocked as well, so we only scored embryos with 14–16 cells. To obtain PAR-6<sup>gut(-)</sup>; *cdc-25.2(0)* embryos, we picked hermaphrodites lacking hT2 and heterozygous for the *cdc-25.2(0)* mutation (balanced by *oxTi980*), and scored embryos (*Figure 4*) and larvae (*Figure 6*) lacking the nuclear GFP expression from *oxTi980*. TBG-1::mCherry marked the MTOC and *end-1*p::GFP::CAAX marked intestinal cell membranes, allowing us to map the position of apical gaps to specific cell positions.

We quantified gaps in these mutant backgrounds in two ways. We examined late comma- to 1.8-fold-stage PAR-6<sup>gut(-)</sup> embryos and treated each cell-cell interface that included at least one star cell as a potential star cell gap site (int1/2, int7/8, and int8/9) and the other cell-cell interfaces as potential non-star cell gap sites (int2/3, int3/4, int4/5, int5/6, int6/7). See *Figure 1—figure supplement 1A* for anatomy. For *cdc-25.2(0)* intestines, we considered int1/2 and int7/8 interfaces as potential star cell gap sites, and the remaining five interfaces (or four interfaces in embryos with 14 cells) as potential non-star cell gap sites. We first counted the number of gaps that affect star cells, and the number of gaps that only affect non-star cells and graphed the proportion of gaps that affect star

vs. non-star cells in *Figure 4F*. Second, we calculated the total number of potential star and non-star cell gap sites and counted the total number of gaps observed at star cell and non-star cell sites, as shown in *Figure 4I*, to understand how the frequency of star cell gaps changed relative to non-star cells. For this analysis, 'PAR-6^gut(-)' embryos were *oxTi980*-carrying sibling embryos from the same experiment. All gaps were scored blind.

### L1 arrest assay

To assess larval viability, 10–20 one-day-old gravid hermaphrodites were picked to an NGM plate and allowed to lay eggs for 2–4 hr at 20°C, with three trials per genotype. Adults were removed, and larvae that grew older than the L1 larval stage were counted and removed over the following three days before counting the number of remaining L1 larvae, which usually appeared unhealthy or dead by day 3.

For the PKC-3 rescue assay strains, gravid hermaphrodites were allowed to lay eggs for up to 12 hr to increase the number of array-positive progeny to analyze. Two independently derived transgenic lines were scored to assess the rescuing capacity of PKC-3(+), PKC-3[G336N], and PKC-3[ΔPB], and one line for PKC-3[K282A].

### Smurf feeding assay

One-day-old gravid hermaphrodites were allowed to lay eggs overnight, and hatched larvae were picked into a 30 μL drop of standard overnight OP50 bacterial culture. 10 μL of a 20% solution of blue food coloring (FD and C Blue #1 Powder, Brand: FLAVORSandCOLOR.com, Amazon) in water was added at a final concentration of 5%, and worms were incubated in a humid box for 3 hr before imaging. To collect the larvae, the dye-OP50 solution was transferred to and spread on an NGM plate to allow the larvae to crawl out of the dark blue solution. Larvae were mounted in 2 mM levamisole on a 3% agarose pad and imaged as described above. Occasionally, larvae were damaged in the transfer process, so larvae that appeared desiccated or damaged were omitted from the analysis. For the PKC-3 rescue assay strains, mosaicism of the array could be detected because transgenically supplied PKC-3 was BFP-tagged and thus visible when it was present or absent. We could not interpret PKC-3 localization as even wild-type PKC-3 appeared cytoplasmic, likely due to its overexpression. However, the presence of BFP allowed us to exclude from our analysis any larvae in which mosaicism appeared to correlate with intestinal defects.

### Statistical analysis

Statistical analyses were performed in Excel and RStudio. When comparing multiple genotypes (*Figures 2D*, *3F*, *4E,* and *7F*; *Figure 7—figure supplement 1G, H*), we used an ANOVA to confirm there were statistically significant differences between groups ($p < 10^{-3}$), followed by a post-hoc Tukey's test for pairwise comparisons of the genotypes to determine p-values in the figures. When comparing one or two genotypes (*Figures 4I*, *5H, J*, *Figure 1—figure supplement 2D–G*), either a Fisher's exact test or a two-tailed t-test was used, with a Bonferroni correction when performing multiple tests.

## Acknowledgements

We thank Jeremy Nance, Dan Dickinson, Ken Kemphues, James McGhee, Alex Lessenger, Ariana Sanchez, Lauren Cote, and Victor Naturale for strains and plasmids, and members of the Feldman lab for helpful discussions about the project and manuscript. Some strains were provided by the CGC, which is funded by NIH Office of Research Infrastructure Programs (P40 OD010440). This work was funded by F32 GM120913-02 and K99 GM13548901 to MDS, F32 GM129900-01 to MAP, and an NIH New Innovator Award DP2 GM119136-01 and R01 GM133950 awarded to JLF.

## Additional information

### Funding

| Funder | Grant reference number | Author |
|---|---|---|
| National Institutes of Health | DP2 GM119136-01 | Jessica L Feldman |
| National Institutes of Health | R01 GM133950 | Jessica L Feldman |
| National Institutes of Health | F32 GM129900-01 | Melissa A Pickett |
| National Institutes of Health | K99 GM13548901 | Maria Danielle Sallee |
| National Institutes of Health | F32 GM120913-02 | Maria Danielle Sallee |
| National Institutes of Health | P40 OD010440 | Jessica L Feldman |

The funders had no role in study design, data collection and interpretation, or the decision to submit the work for publication.

### Author contributions

Maria Danielle Sallee, Conceptualization, Resources, Formal analysis, Funding acquisition, Validation, Investigation, Visualization, Methodology, Writing - original draft, Writing - review and editing; Melissa A Pickett, Conceptualization, Resources, Writing - review and editing; Jessica L Feldman, Conceptualization, Resources, Supervision, Funding acquisition, Writing - original draft, Project administration, Writing - review and editing

### Author ORCIDs

Maria Danielle Sallee (iD) https://orcid.org/0000-0002-0144-2607
Jessica L Feldman (iD) https://orcid.org/0000-0002-5210-5045

### Decision letter and Author response

Decision letter https://doi.org/10.7554/eLife.64437.sa1
Author response https://doi.org/10.7554/eLife.64437.sa2

## Additional files

### Supplementary files

- Supplementary file 1. CRISPR allele primers and plasmids.

- Transparent reporting form

### Data availability

All data generated or analysed during this study are included in the manuscript and supporting files. Source data files have been provided for Figures 6 and 7.

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
