## [Decision Letter]

**Acceptance summary:**

This paper successfully describes how PAR proteins are needed after polarity establishment to build and maintain a continuous intestinal lumen as cells undergo division and elongation. The ability to spatially and temporally target PAR protein degradation is an important technical advance that provides convincing evidence for the PAR complex being needed for apical maintenance during morphogenesis. The findings will be of broad interest to developmental and cell biologists, as well as experts in cell polarity.

**Decision letter after peer review:**

Thank you for submitting your article "Apical PAR complex proteins protect against epithelial assaults to create a continuous and functional intestinal lumen" for consideration by *eLife*. Your article has been reviewed by three peer reviewers, one of whom is a member of our Board of Reviewing Editors, and the evaluation has been overseen by Anna Akhmanova as the Senior Editor. The reviewers have opted to remain anonymous.

The reviewers have discussed the reviews with one another and the Reviewing Editor has drafted this decision to help you prepare a revised submission.

Summary:

This paper provides insight into how apicobasal polarity is necessary for the developing intestinal epithelium to form during morphogenesis. While the importance of polarity in gut development has been known, this study uses a new tissue-specific degradation strategy to probe the effects of polarity loss during cell division and tissue elongation during gut development in *C. elegans*. Live imaging experiments show that dividing cells maintain apical PAR complex proteins, but lose apical MTOC proteins, during mitosis. The apical domain, which is retained through mitosis, is used to rebuild the lost apical MTOC after division. Degradation of Par6, aPKC and Cdc42 cause similar morphogenetic defects in the intestine, leading to apical gaps in the intestinal lumen, gut obstruction during feeding, and larval lethality. These phenotypes occur most prominently in cells that undergo mitosis and during gut elongation, highlighting the continued need for PAR function in cells as they undergo morphogenetic changes. Rescue experiments with mutant forms of PKC-3/aPKC suggest a novel, kinase-independent role for PKC-3 in intestinal development.

The most significant advance of the study is the suggestion that PAR-6 and PKC-3 function to mark the apical surface in dividing cells, allowing apical MTs to reform once division is complete. In addition, the authors propose that PAR-6 and PKC-3 function to maintain apical junctions as cells change shape during the process of embryo elongation. Overall, the study is rigorous and convincing, although as detailed below, additional experiments and data quantitation are needed to test the main advances of the paper.

Essential revisions:

Revisions requiring new experiments

1. The hypothesis that PAR-6 and PKC-3 mark the apical domain so that it can be recognized after mitosis is complete, allowing for reassembly of apical MTs, is intriguing, but additional experiments are needed to more thoroughly test this idea. Using heat-shock expression of ZIF-1, it should be possible to degrade these proteins at specific points of intestinal tube formation to test key hypotheses. PAR-6 or PKC-3 should be depleted just after star cell division is complete to ensure that the MT gaps that form at apical surfaces of star cells when PAR-6 is depleted early do not appear. Conversely, if PAR-6 is depleted early then re-expressed by heat-shock or a late-acting promoter after star cell division is complete, do the star cell gaps disappear and fill once again with apical MTs or do gaps remain? I don't think these experiments need to be done with tissue-specific degradation since the autonomy of phenotypes has already been established. Although it's possible that the timing is too tight to get this to work, the authors are encouraged to try this experiment as they would know quickly whether it would be feasible, and it would strengthen their conclusions.

2. A second and related question is whether PAR-6 and PKC-3 actually have a role during elongation itself. Two alternative hypotheses are that gaps take time to develop and would form eventually even in embryos that do not elongate; or that small gaps present initially are not revealed until apical domain area expands as cells change shape during elongation. The experiment above to remove PAR-6 or PKC-3 after star cell division but before or during elongation can be used to separate out an early function in junction formation from a later function during elongation. Alternatively (or additionally), elongation could be blocked in embryos lacking PAR-6 or PKC-3 in gut cells to determine if defects still arise.

Revisions requiring new data analysis

Figure 1: The data showing protein localization during star cell division would benefit from a temporal element: images of protein localization prior the onset of mitosis (or alternatively, images of a neighboring non-mitotic cell), should be shown to illustrate that proteins of interest localize apically during interphase and are then lost during division. A kymograph from a line scan across the apical membrane might be one useful way to show how the localization of these proteins change over time.

Figure 1F: Related to the previous point, the DLG-1 and HMR-1 localizations are particularly difficult to interpret based on the images shown. Images of these proteins before, during and after mitosis would be helpful, as would quantification.

Figure 5G-I: Quantification of basolateral protein levels at the midline is needed to show that they are mislocalized – the authors should measure intensities or make line scans rather than just classifying embryos as "Line/Hazy/Absent," which are subjective terms.

Figure 8L: Because the PAR-6 mutant worms lack a clear midline, the quantification of worms with LGL-1 at the midline is difficult to verify. Given the profound defects in tissue structure, my suspicion is that quantification of these data is more likely to be confusing than helpful. Perhaps it would be preferable in this case to simply show the images and state in the text that LGL-1 appears to be present on all membranes.

**Author response:**

*Essential revisions:*

*Revisions requiring new experiments*

*1. The hypothesis that PAR-6 and PKC-3 mark the apical domain so that it can be recognized after mitosis is complete, allowing for reassembly of apical MTs, is intriguing, but additional experiments are needed to more thoroughly test this idea. Using heat-shock expression of ZIF-1, it should be possible to degrade these proteins at specific points of intestinal tube formation to test key hypotheses. PAR-6 or PKC-3 should be depleted just after star cell division is complete to ensure that the MT gaps that form at apical surfaces of star cells when PAR-6 is depleted early do not appear. Conversely, if PAR-6 is depleted early then re-expressed by heat-shock or a late-acting promoter after star cell division is complete, do the star cell gaps disappear and fill once again with apical MTs or do gaps remain? I don't think these experiments need to be done with tissue-specific degradation since the autonomy of phenotypes has already been established. Although it's possible that the timing is too tight to get this to work, the authors are encouraged to try this experiment as they would know quickly whether it would be feasible, and it would strengthen their conclusions.*

We thank the reviewers for suggesting these important experiments. We used three approaches to further bolster the model that PAR-6 is required to maintain apical continuity through epithelial divisions:

1) Alleviate the requirement for PAR-6 in the star cells by blocking star cell divisions: We observed frequent gaps in PAR-6^gut(-)^ embryos at the anterior and posterior ends of the intestine, positions where star cells divided. To determine if cell division drives gap formation, we genetically blocked star cell divisions (*cdc-25.2(0)*) in PAR-6^gut(-)^ embryos and found a significant reduction in gap formation at the intestine ends (new in Figure 4), indicating that divisions do indeed contribute to gap formation. In addition, when we compared the position of intestinal obstructions in larvae, we found that 25% of PAR-6^gut(-)^*; cdc-25.2(0)* intestines had a continuous int1/int2 connection and a more posterior obstruction (new in Figure 6), compared to only 5% in PAR-6^gut(-)^ larvae, suggesting that cell division contributes to the high frequency of intestinal obstructions at int1.

2) Remove PAR-6 from the intestine after the star cell divisions: The *asp-1* promoter drives intestinal expression (Tcherepanova et al. 2000) after the star cell divisions, beginning around the 2-fold stage. We made and expressed an *asp-1p::zif-1* transgene and observed its ability to robustly degraded PAR-6 after the star cell divisions at the two-fold embryonic stage (new Figure 6—figure supplement 1). 16% of these “PAR-6^late gut(-)^” larvae arrest in the L1 stage and have obstructed intestines, but none had an int1 obstruction, suggesting that removing PAR-6 after the star cell divisions allows a normal int1/int2 connection to form.

3) Modulate expression timing using heat shock: As suggested, we attempted to express ZIF-1 immediately after the star cell divisions using a heat-shock driven promoter (*hsp-1*p). We generated *hsp-1*p*::zif-1* transgenes, but unfortunately, these transgenes were too variable and leaky to interpret the results of these experiments with confidence.

Together, we believe that these new experiments strengthen our conclusion that PAR-6 is required to maintain MTOC continuity through cell divisions, and that cell divisions are an obstacle to apical continuity.

*2. A second and related question is whether PAR-6 and PKC-3 actually have a role during elongation itself. Two alternative hypotheses are that gaps take time to develop and would form eventually even in embryos that do not elongate; or that small gaps present initially are not revealed until apical domain area expands as cells change shape during elongation. The experiment above to remove PAR-6 or PKC-3 after star cell division but before or during elongation can be used to separate out an early function in junction formation from a later function during elongation. Alternatively (or additionally), elongation could be blocked in embryos lacking PAR-6 or PKC-3 in gut cells to determine if defects still arise.*

We used the reagents introduced above to assess whether PAR-6 is required in later elongating embryonic intestines, or only in the early stages. Using the *asp-1*p::*zif-1* transgene, we depleted PAR-6 in the 2-fold embryonic stage after the star cell divisions were completed (new Figure 6—figure supplement 1). As mentioned in the previous section, we observed L1 arrest and intestinal obstructions in elongating non-dividing cells in 16% of L1 larvae. This result suggests that PAR-6 continues to be required during embryonic intestinal elongation, and not only in the early stages of intestinal development. The observed defects were much milder than those caused by early depletion of PAR-6, however, which suggests that PAR-6 is required both during early and later stages of embryonic intestinal development.

We tried to block intestinal elongation to distinguish the time vs. elongation hypotheses.

We avoided the classic approach of inhibiting actin to block elongation (Priess and Hirsch 1986), because of the newly identified role for actin in anchoring microtubules in the embryonic intestine (Sanchez et al., 2021). Instead, we simultaneously degraded PAR-6 in both the intestine and in the skin. PAR-6 degradation in the skin was achieved using an “*epiDeg”* transgene (Wang et al., 2017) that would target the endogenous ZF-tagged PAR-6 in the skin and thereby block embryonic elongation (Totong, Achilleos and Nance, 2007). Unfortunately, the degree to which embryos arrested was highly variable, and embryo movement (an important criterion to ensure we were analyzing live embryos) introduced uncertainty in intestinal length measurements, particularly as embryos often stretched and contracted along their AP axis, causing the apparent length of the intestine to change along with it. We observed some arrested embryos with gaps becoming worse over the course of 1 hour and others with gaps remaining constant, but the variability and uncertainty in the degree of intestinal elongation arrest made us unable to confidently interpret the phenotypes, so we did not include this analysis in the revised manuscript.

*Revisions requiring new data analysis*

*Figure 1: The data showing protein localization during star cell division would benefit from a temporal element: images of protein localization prior the onset of mitosis (or alternatively, images of a neighboring non-mitotic cell), should be shown to illustrate that proteins of interest localize apically during interphase and are then lost during division. A kymograph from a line scan across the apical membrane might be one useful way to show how the localization of these proteins change over time.*

We thank the reviewers for this suggestion which we have address in two ways. First, we have re-cropped the images in Figure 1 to show both the dividing star cells and their non-dividing posteriors neighbors for comparison. Second, we have added Figure 1—figure supplement 2 to show representative MTOC, apical, and junctional protein localization before and during the star cell divisions, as well as quantifications of each cell type and time point.

*Figure 1F: Related to the previous point, the DLG-1 and HMR-1 localizations are particularly difficult to interpret based on the images shown. Images of these proteins before, during and after mitosis would be helpful, as would quantification.*

We agree that the localizations of junction proteins are difficult to visualize during divisions. We have added additional images and analysis of HMR-1 and DLG-1 intensity before and during mitosis in Figure 1—figure supplement 2B, 2D, and 2E.

*Figure 5G-I: Quantification of basolateral protein levels at the midline is needed to show that they are mislocalized – the authors should measure intensities or make line scans rather than just classifying embryos as "Line/Hazy/Absent," which are subjective terms.*

In place of our original analysis in Figure 5, we have added quantification of the intensity of basolateral markers LET-413 and LGL-1, as well as the apical markers PAR-3 and TBG-1, at apical and lateral surfaces (Figure 5H, J).

*Figure 8L: Because the PAR-6 mutant worms lack a clear midline, the quantification of worms with LGL-1 at the midline is difficult to verify. Given the profound defects in tissue structure, my suspicion is that quantification of these data is more likely to be confusing than helpful. Perhaps it would be preferable in this case to simply show the images and state in the text that LGL-1 appears to be present on all membranes.*

Without the benefit of an independent marker, we were not able to quantify LGL-1 intensity at different intestinal membranes. We have kept our original images and edited the text to make the point clearer.

---

## [Author Response]

Essential revisions:Revisions requiring new experiments1. The hypothesis that PAR-6 and PKC-3 mark the apical domain so that it can be recognized after mitosis is complete, allowing for reassembly of apical MTs, is intriguing, but additional experiments are needed to more thoroughly test this idea. Using heat-shock expression of ZIF-1, it should be possible to degrade these proteins at specific points of intestinal tube formation to test key hypotheses. PAR-6 or PKC-3 should be depleted just after star cell division is complete to ensure that the MT gaps that form at apical surfaces of star cells when PAR-6 is depleted early do not appear. Conversely, if PAR-6 is depleted early then re-expressed by heat-shock or a late-acting promoter after star cell division is complete, do the star cell gaps disappear and fill once again with apical MTs or do gaps remain? I don't think these experiments need to be done with tissue-specific degradation since the autonomy of phenotypes has already been established. Although it's possible that the timing is too tight to get this to work, the authors are encouraged to try this experiment as they would know quickly whether it would be feasible, and it would strengthen their conclusions.

We thank the reviewers for suggesting these important experiments. We used three approaches to further bolster the model that PAR-6 is required to maintain apical continuity through epithelial divisions:

1) Alleviate the requirement for PAR-6 in the star cells by blocking star cell divisions: We observed frequent gaps in PAR-6^gut(-)^ embryos at the anterior and posterior ends of the intestine, positions where star cells divided. To determine if cell division drives gap formation, we genetically blocked star cell divisions (*cdc-25.2(0)*) in PAR-6^gut(-)^ embryos and found a significant reduction in gap formation at the intestine ends (new in Figure 4), indicating that divisions do indeed contribute to gap formation. In addition, when we compared the position of intestinal obstructions in larvae, we found that 25% of PAR-6^gut(-)^*; cdc-25.2(0)* intestines had a continuous int1/int2 connection and a more posterior obstruction (new in Figure 6), compared to only 5% in PAR-6^gut(-)^ larvae, suggesting that cell division contributes to the high frequency of intestinal obstructions at int1.

2) Remove PAR-6 from the intestine after the star cell divisions: The *asp-1* promoter drives intestinal expression (Tcherepanova et al. 2000) after the star cell divisions, beginning around the 2-fold stage. We made and expressed an *asp-1p::zif-1* transgene and observed its ability to robustly degraded PAR-6 after the star cell divisions at the two-fold embryonic stage (new Figure 6—figure supplement 1). 16% of these “PAR-6^late gut(-)^” larvae arrest in the L1 stage and have obstructed intestines, but none had an int1 obstruction, suggesting that removing PAR-6 after the star cell divisions allows a normal int1/int2 connection to form.

3) Modulate expression timing using heat shock: As suggested, we attempted to express ZIF-1 immediately after the star cell divisions using a heat-shock driven promoter (*hsp-1*p). We generated *hsp-1*p*::zif-1* transgenes, but unfortunately, these transgenes were too variable and leaky to interpret the results of these experiments with confidence.

Together, we believe that these new experiments strengthen our conclusion that PAR-6 is required to maintain MTOC continuity through cell divisions, and that cell divisions are an obstacle to apical continuity.

2. A second and related question is whether PAR-6 and PKC-3 actually have a role during elongation itself. Two alternative hypotheses are that gaps take time to develop and would form eventually even in embryos that do not elongate; or that small gaps present initially are not revealed until apical domain area expands as cells change shape during elongation. The experiment above to remove PAR-6 or PKC-3 after star cell division but before or during elongation can be used to separate out an early function in junction formation from a later function during elongation. Alternatively (or additionally), elongation could be blocked in embryos lacking PAR-6 or PKC-3 in gut cells to determine if defects still arise.

We used the reagents introduced above to assess whether PAR-6 is required in later elongating embryonic intestines, or only in the early stages. Using the *asp-1*p::*zif-1* transgene, we depleted PAR-6 in the 2-fold embryonic stage after the star cell divisions were completed (new Figure 6—figure supplement 1). As mentioned in the previous section, we observed L1 arrest and intestinal obstructions in elongating non-dividing cells in 16% of L1 larvae. This result suggests that PAR-6 continues to be required during embryonic intestinal elongation, and not only in the early stages of intestinal development. The observed defects were much milder than those caused by early depletion of PAR-6, however, which suggests that PAR-6 is required both during early and later stages of embryonic intestinal development.

We tried to block intestinal elongation to distinguish the time vs. elongation hypotheses.

We avoided the classic approach of inhibiting actin to block elongation (Priess and Hirsch 1986), because of the newly identified role for actin in anchoring microtubules in the embryonic intestine (Sanchez et al., 2021). Instead, we simultaneously degraded PAR-6 in both the intestine and in the skin. PAR-6 degradation in the skin was achieved using an “*epiDeg”* transgene (Wang et al., 2017) that would target the endogenous ZF-tagged PAR-6 in the skin and thereby block embryonic elongation (Totong, Achilleos and Nance, 2007). Unfortunately, the degree to which embryos arrested was highly variable, and embryo movement (an important criterion to ensure we were analyzing live embryos) introduced uncertainty in intestinal length measurements, particularly as embryos often stretched and contracted along their AP axis, causing the apparent length of the intestine to change along with it. We observed some arrested embryos with gaps becoming worse over the course of 1 hour and others with gaps remaining constant, but the variability and uncertainty in the degree of intestinal elongation arrest made us unable to confidently interpret the phenotypes, so we did not include this analysis in the revised manuscript.

Revisions requiring new data analysisFigure 1: The data showing protein localization during star cell division would benefit from a temporal element: images of protein localization prior the onset of mitosis (or alternatively, images of a neighboring non-mitotic cell), should be shown to illustrate that proteins of interest localize apically during interphase and are then lost during division. A kymograph from a line scan across the apical membrane might be one useful way to show how the localization of these proteins change over time.

We thank the reviewers for this suggestion which we have address in two ways. First, we have re-cropped the images in Figure 1 to show both the dividing star cells and their non-dividing posteriors neighbors for comparison. Second, we have added Figure 1—figure supplement 2 to show representative MTOC, apical, and junctional protein localization before and during the star cell divisions, as well as quantifications of each cell type and time point.

Figure 1F: Related to the previous point, the DLG-1 and HMR-1 localizations are particularly difficult to interpret based on the images shown. Images of these proteins before, during and after mitosis would be helpful, as would quantification.

We agree that the localizations of junction proteins are difficult to visualize during divisions. We have added additional images and analysis of HMR-1 and DLG-1 intensity before and during mitosis in Figure 1—figure supplement 2B, 2D, and 2E.

Figure 5G-I: Quantification of basolateral protein levels at the midline is needed to show that they are mislocalized – the authors should measure intensities or make line scans rather than just classifying embryos as "Line/Hazy/Absent," which are subjective terms.

In place of our original analysis in Figure 5, we have added quantification of the intensity of basolateral markers LET-413 and LGL-1, as well as the apical markers PAR-3 and TBG-1, at apical and lateral surfaces (Figure 5H, J).

Figure 8L: Because the PAR-6 mutant worms lack a clear midline, the quantification of worms with LGL-1 at the midline is difficult to verify. Given the profound defects in tissue structure, my suspicion is that quantification of these data is more likely to be confusing than helpful. Perhaps it would be preferable in this case to simply show the images and state in the text that LGL-1 appears to be present on all membranes.

Without the benefit of an independent marker, we were not able to quantify LGL-1 intensity at different intestinal membranes. We have kept our original images and edited the text to make the point clearer.